# WHO GETS CITED MOST?
# BENCHMARKING LONG-CONTEXT LANGUAGE MODELS ON SCIENTIFIC ARTICLES

## ABSTRACT

This paper introduces SciTrek, a novel question-answering benchmark designed to evaluate the long-context reasoning capabilities of large language models (LLMs) using scientific articles. Current long-context benchmarks often rely on non-scientific texts, focus on simple information retrieval tasks, or employ artificial contexts. SciTrek addresses these limitations by proposing complex questions that require information aggregation and synthesis across multiple full-text scientific articles. Questions and their ground-truth answers are automatically generated by formulating them as SQL queries over a database constructed from article metadata (titles, authors, and references). The SQL operations provide explicit, verifiable reasoning steps for fine-grained error analysis, and the construction process scales to contexts up to 1M tokens with minimal supervision. Extensive experiments on a diverse set of open-weight and proprietary LLMs demonstrate that SciTrek poses a significant challenge as the context length increases, with supervised fine-tuning and reinforcement learning offering only limited gains. Our analysis reveals systematic shortcomings in models' abilities to perform basic numerical operations and accurately locate specific information in long contexts.[1]

## 1 INTRODUCTION

Large language models (LLMs) show promise in accelerating scientific progress by assisting researchers in reviewing and synthesizing the growing body of literature (Pearson, 2024; Agarwal et al., 2025) and generating novel research ideas (Si et al., 2025; Baek et al., 2025). So much so that industry products such as Deep Research[2], Elicit[3], and Scite[4] have emerged as dedicated tools to aid users with complex, multi-step research tasks. Scientific workflows often require processing large inputs: full scientific articles, collections of references, or structured datasets. A model capable of processing 100K+ tokens could simultaneously analyze multiple articles, track long chains of reasoning, or connect experimental results to prior work.

While long-context language models (LCLMs) are highly relevant to scientific tasks, existing benchmarks designed to evaluate their capabilities predominantly focus on non-scientific texts (Kuratov et al., 2024; Maekawa et al., 2025; Bai et al., 2025; Yen et al., 2025). They mostly address simple information retrieval tasks (e.g., Needle-In-A-Haystack; Kamradt 2023; Hsieh et al. 2024) rather than information integration, i.e., reasoning across multiple documents to synthesize information and answer complex questions. Some question-answering benchmarks (Asai et al., 2024; An et al., 2024; Bai et al., 2025) require information integration, but do not provide a typology of the reasoning steps involved, which makes it difficult to determine the specific capabilities being assessed and why a model fails. Another issue relates to scalability and extensibility. When substantial human effort is involved in generating the questions and their answers (e.g., Asai et al. 2024), scaling to longer contexts or different types of questions becomes prohibitive. To better emulate scientific research practice, it is also important to evaluate a range of skills in natural contexts, such as a model's

---

[1]Our dataset can be downloaded from `xxx.yyy.zzz`

[2]`https://openai.com/index/introducing-deep-research/`

[3]`https://elicit.com/`

[4]`https://scite.ai/`

| Question | SQL Query | Answer |
|---|---|---|
| What is the highest number of authors that any single article has? | `SELECT MAX(author_count) FROM articles` | 10 |
| What is the word count of the titles of articles, sorted by the number of authors in ascending order? | `SELECT title_word_count FROM articles ORDER BY author_count ASC` | 9, 17, 5, 6, 9, 12 |
| How many references do articles with exactly two authors have? | `SELECT reference_count FROM articles WHERE author_count = 2` | 16 |
| What is the total number of words in the titles of all articles that have exactly 60 references? | `SELECT SUM(title_word_count) FROM articles WHERE reference_count = 60` | 13 |
| What are the names of the authors who are either the first or second author of an article, listed in descending order of their position? | `SELECT author_name FROM article_author WHERE author_position < 2 ORDER BY author_position DESC` | N. Shazeer, A. Vaswani |
| How many articles have been cited by other articles but do not cite any other articles? | `SELECT COUNT(*) FROM articles WHERE article_id NOT IN (SELECT article_id_citing FROM citing_cited) AND article_id IN (SELECT article_id_cited FROM citing_cited)` | 2 |

Table 1: The SciTrek benchmark: example questions with corresponding SQL queries and answers.

ability to aggregate or filter information or combinations thereof (Wang et al., 2024c). However, a few benchmarks rely on artificial contexts, often created by extending short-context tasks (e.g., Hot-potQA; Yang et al. 2018) with added noise (Kuratov et al., 2024) or by synthesizing the input with LLMs (Maekawa et al., 2025).

In this paper, we create a question-answering benchmark based on scientific articles that aims to alleviate these issues. We assume a scenario where a researcher, possibly during the literature review process, seeks answers to specific questions when the related scientific articles are provided. As an example, consider the first question in Table 1: "*What is the highest number of authors that any single article has?*". Although seemingly simple, it requires aggregation of information across multiple articles. To answer correctly, a hypothetical model would need to count the number of authors for each article within a document collection, and identify the article with the largest number. To generate questions demanding explicit reasoning with natural language inputs, we focus on basic numerical operations such as counting, aggregating, and sorting, applied to key elements of scientific articles: titles, authors, and references. Since these operations correspond to standard database functions, we can automatically generate arbitrarily complex SQL queries over any collection of scientific articles and obtain ground-truth answers without manual annotation by constructing database tables for titles, authors, and references. We take advantage of recent advances in LLM capabilities for SQL understanding and generation (Yu et al., 2018; Li et al., 2023; Hui et al., 2024) to automatically convert SQL queries into corresponding natural language questions.

Our benchmark, SciTrek, contains a test set of 2,121 question-answer pairs on scientific articles and a training set of 19,543 question-answer pairs, with contexts of varying lengths (i.e., 64K, 128K, 512K, and 1M[5]). Table 1 illustrates examples of questions, their corresponding SQL queries, and answers. These questions are relatively superficial, as they do not engage deeply with article content. However, the metadata consists of distillable facts sparsely distributed across article collections, requiring models to retrieve and synthesize information over long contexts. If models struggle with simple comparing, filtering, sorting and aggregating, they are unlikely to succeed on more complex analytical tasks (Wolfson et al., 2020; 2025).[6] We use SciTrek to evaluate the capabilities of LCLMs with context windows exceeding 128K tokens. Experimental results reveal that the benchmark poses significant challenges, with models achieving poor performance, even when enhanced with supervised fine-tuning (SFT) and reinforcement learning (RL). We make the following contributions:

- We propose a methodology for constructing a long-context question-answering benchmark over natural texts, with explicit reasoning processes approximated via SQL operations that involve in-

---

[5]Throughout the paper, K refers to 1,024 and M to 1,024K.

[6]Appendix A contains expert-written example questions on scientific articles, which likewise call for these capabilities.

| Dataset | Natural Context | #SciQs | Reasoning | Scalable | Len | Example Question |
|---|---|---|---|---|---|---|
| NeedleBench | ✗ | — | ✗ | ✓ | 128K | What legendary item is hidden on Emerald Island? |
| Ada-LEval | ✗ | — | ✗ | ✓ | 128K | What is the correct order of the segments? |
| BABILong | ✗ | — | ✗ | ✓ | 10M | Where is Mary? |
| HELMET | ✗ | — | ✗ | ✗ | 128K | Who set the fire in one tree hill? |
| LIFBENCH | ✗ | — | ✗ | ✓ | 128K | Retrieve the entry at position 8th in the list. |
| RULER | ✗ | — | ✗ | ✓ | 128K | Find all variables that are assigned the value 12345. |
| OpenScholar | ✓ | 208 | ✗ | ✗ | 3K* | Compile a list of reviews [...], and identify the most promising [...] |
| LongBench v2 | ✓ | 50 | ✗ | ✗ | 128K | How long have I been living in my current apartment in Shinjuku? |
| LongMemEval | ✓ | — | ✗ | ✗ | 2M | How many bikes do I currently own? |
| L-Eval | ✓ | — | ✗ | ✗ | 200K | How do I know when I should apply for Medicare? |
| HoloBench | ✗ | — | ✓ | ✓ | 64K | What are the names of wines and their corresponding grape types? |
| MathHay | ✗ | — | ✓ | ✓ | 128K | What is the total number of points scored by LeBron [...] combined? |
| Loong | ✓ | 53 | ✓ | ✓ | 250K | Which company has the highest non-current assets? |
| SciTrek | ✓ | 1,716 | ✓ | ✓ | 1M | How many articles have been cited by other articles but do not cite any other articles? |

Table 2: Representative benchmarks for evaluating LCLMs. Natural Context: does the benchmark have natural input contexts; #SciQs: the number of unique benchmark questions or instructions with gold answers on scientific articles (—: benchmarks that are not using any scientific articles); Reasoning: does the benchmark provide detailed reasoning skills required to answer each question; Scalable: can the benchmark be extended with minimal effort (e.g., to longer contexts or larger datasets); Len: the maximum input length that the model supports in terms of tokens; *: length based on texts retrieved rather than full input (which is not available).

formation aggregation. The methodology requires minimal human intervention and can be replicated for other domains containing entities that can be structured as database tables.

- Extensive experiments reveal that frontier open-weight and proprietary LLMs struggle significantly with this task, especially as input length increases. While SFT and RL improve over baseline models, performance remains limited. All models struggle with questions involving references and citation relationships among the input articles.

- By design, SciTrek enables fine-grained analysis of model behavior, offering insight into where and why models fail. Our evaluation of open-weight models reveals systematic shortcomings in counting bibliographic elements and performing basic numerical operations, with error rates increasing as question length grows. Interestingly, models frequently misinterpret compound conditions and struggle with logical constructs involving negation. (see the last question in Table 1).

## 2 RELATED WORK

**Long-context Language Modeling** A growing body of work explores the challenges LLMs encounter when processing inputs that exceed their context lengths. Rotary Position Embeddings (RoPE; Su et al. 2024) have emerged as a widely used approach, replacing absolute positional embeddings with rotational transformations of token embeddings. It improves generalization to unseen sequence lengths. Building on RoPE, several extensions introduce rescaling to further enhance performance on substantially longer inputs without the need for retraining (Peng et al., 2024).

Other architectural advances focus on reducing the quadratic cost of the Transformer attention mechanism (Sun et al., 2025). Sparse attention methods, such as LongFormer (Beltagy et al., 2020),

compute only selected portions of the full attention matrix, thereby reducing computational cost and accelerating both prefilling and inference (Jiang et al., 2024; Fu et al., 2024). Many contemporary models interleave sparse and full attention layers to balance efficiency and performance (Dubey et al., 2024; Llama Team, 2025; Yang et al., 2025; Kamath et al., 2025). Another popular strategy of reducing the computation required in long-context settings is to limit the number of active model parameters. Group Query Attention (GQA; Ainslie et al. 2023) enables multiple attention heads to share key–value projections, without significant performance loss. The open-weight LLMs used in our experiments adopt combinations of these techniques, complemented by pretraining on long-context tasks (Dubey et al., 2024; Kamath et al., 2025; Yang et al., 2025; Llama Team, 2025).

**Long-context Datasets** With growing interest in LCLMs, there has been a surge in benchmarks designed to evaluate their performance. As far as text-based efforts are concerned, initial benchmarks primarily assessed whether models are able to retrieve relevant information from their context, typically through Needle-In-A-Haystack (NIAH) tasks, such as NeedleBench (Li et al., 2024). Building on these, subsequent benchmarks introduced either distractor information to short-context tasks or dispersed relevant information across synthetic contexts, such as Ada-LEval (Wang et al., 2024a), BABILong (Kuratov et al., 2024), HELMET (Yen et al., 2025), LIFBENCH (Wu et al., 2025b), LongBench (Bai et al., 2024), and RULER (Hsieh et al., 2024).

In addition to retrieval, some benchmarks emphasize complex reasoning over long contexts. For example, OpenScholar (Asai et al., 2024) and LongBench v2 (Bai et al., 2025) contain questions that require extensive use of the multiple documents provided as context, while DocFinQA (Reddy et al., 2024) and MedOdyssey (Fan et al., 2025) focus on financial and medical reasoning, respectively. Although not explicitly technical, LongMemEval (Wu et al., 2025a) and L-Eval (An et al., 2024) also requires reasoning over natural, long-context information. However, evaluating model answers against these benchmarks remains challenging, or the reasoning steps required to arrive at correct responses are often opaque, making it difficult to assess failure modes of LCLMs. In contrast, HoloBench (Maekawa et al., 2025) and MathHay (Wang et al., 2024b) better delineate between the types of reasoning required for each question, but rely on unnatural contexts. Different from Loong (Wang et al., 2024c) which also combines structured reasoning and natural contexts, SciTrek covers more unique benchmark questions on scientific articles. CURIE (Cui et al., 2025), OpenScholar and LongBench v2 evaluate reasoning over scientific articles, but their reliance on expert-curated questions and answers poses challenges for scalability to longer contexts or larger datasets.

In designing SciTrek, we aim to retain many of the useful qualities of previous benchmarks (e.g., structured reasoning, scalability) while still proposing a fairly natural and easy task for humans to perform. Although we focus on scientific articles, our methodology extends to other domains with explicit entities and relations. Thanks to the SQL backbone, we are able to construct questions testing various model skills (e.g., aggregation, filtering), and how these manifest themselves through explicit reasoning processes. We compare SciTrek with representative existing benchmarks in Table 2.

## 3 THE SCITREK BENCHMARK

In this section, we explain how SciTrek was curated. As mentioned earlier, our benchmark consists of question-answer pairs over scientific articles. Figure 1 illustrates our process: we first gather scientific article collections as contexts corresponding to different lengths (i.e., 64K, 128K, 512K, and 1M); we then create databases representing the article metadata and SQL queries with answers from database execution; finally, we convert the SQL queries into natural language questions.

### 3.1 GATHERING SCIENTIFIC ARTICLE COLLECTIONS

SciTrek is constructed using scientific articles from Semantic Scholar.[7] To cover diverse topics of articles, we obtain an initial set of seed articles from eight subjects: Computer Science (CS), Economics, Electronic Engineering (EE), Math, Physics, Biology, Finance, and Statistics.[8] For each subject, we select two seed articles with more than 100 citations since 2020. For each article,

---

[7]Semantic Scholar: https://www.semanticscholar.org/

[8]Following the typology of subjects from https://arxiv.org/.

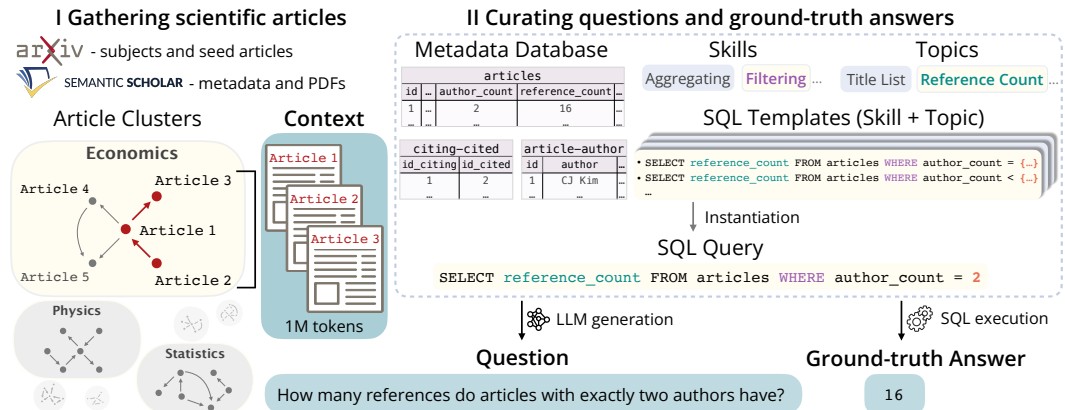

Figure 1: Overview of SciTrek construction process: we gather article collections of varying scales; we then obtain SQL queries and their answers based on databases (which store collection-specific metadata); and finally we convert SQL queries to natural language questions. The dataset consists of input full-text contexts, questions and their answers, highlighted with a blue background.

we retrieve related articles from Semantic Scholar.[9] To ensure broad coverage, we include two-hop related articles identified via Semantic Scholar's citation graph based on the reference and citation list. For each seed article, we randomly sample ten first-hop related articles, and for each of these, we further sample five second-hop related articles to form an article cluster. Since full texts are required, we filter out articles without PDFs. This process yields 16 article clusters comprising 662 scientific articles with PDFs across eight subjects (see step I in Figure 1). Finally, we convert the collected PDFs into markdown texts using Marker.[10]

From these clusters, we construct article collections of varying context lengths for our question-answering task. By concatenating the markdown texts, we generate collections with lengths of 64K, 128K, 512K, and 1M tokens.[11] Each collection is initialized with a randomly selected article, and additional articles are added until it reaches a specified length level. We expand the collections using two strategies: (1) random sampling from the clusters, and (2) traversing the citation graph within each cluster (using both depth-first and breadth-first search) to construct collections that preserve citation relations among articles. Each collection contains at least four articles, and no two collections share more than half of their articles. From the 662 scientific articles across 16 clusters, we construct 2,612 article collections spanning all four length levels, including 2,027 generated through random sampling and 585 through graph traversal.

## 3.2 CREATING DATABASES AND SQL QUERIES

Once the article collections are assembled, we construct a database for each (see step II in Figure 1). We focus on a core subset of elements shared across all scientific articles: titles, authors, and references. Based on these key elements, each database comprises three tables: *articles*, *article-author*, and *citing-cited*. The *articles* table contains metadata such as the title, reference count, and title word count. The *article-author* table captures author information, including names and their positions in the corresponding author list. The *citing-cited* table records citation relations among articles. A detailed description of these tables is provided in Appendix B. This information is obtained from Semantic Scholar or derived via simple preprocessing, such as splitting titles and counting words.

---

[9]Semantic Scholar provides APIs to retrieve scientific articles with metadata including titles, authors, a reference list with articles that the current article cites, and a citation list with articles that cite the current article. Throughout this paper, a reference is a bibliographic entry listed in the reference section of a scientific article, whereas a citation is an in-text acknowledgement of another's work within the article.

[10]https://github.com/datalab-to/marker

[11]We can easily construct collections with arbitrary lengths exceeding 1M tokens.

| SQL Commands | |
| --- | --- |
| Aggregating | MAX, MIN, SUM, AVG, COUNT, DISTINCT |
| Sorting | ORDER BY, ASC, DESC, GROUP BY |
| Filtering | WHERE |

| SQL Operators | |
| --- | --- |
| Comparison | =, >, <, >=, <=, <>, LIKE |
| Arithmetic | +, −, ∗, /, % |
| Logical | AND, NOT, OR, BETWEEN, IN |

Table 3: Core commands and operators in SQL.

| Skill | Count | Example Query Template |
| --- | --- | --- |
| Aggregating | 20 | SELECT MAX(author_count) FROM articles |
| Sorting | 27 | SELECT title_word_count FROM articles ORDER BY author_count ASC |
| Filtering | 107 | SELECT author_name FROM article_author WHERE author_position = {author-position} |
| Filtering+ Aggregating | 107 | SELECT SUM(title_word_count) FROM articles WHERE reference_count = {reference-count} |
| Filtering+ Sorting | 106 | SELECT author_count FROM articles WHERE title_word_count % 2 = 1 ORDER BY title_word_count DESC |
| Relational Filtering | 20 | SELECT COUNT(∗) FROM articles WHERE article_id NOT IN (SELECT article_id_citing FROM citing_cited) AND article_id IN (SELECT article_id_cited FROM citing_cited) |

Table 4: SQL templates representing different information processing skills. {author-position}, {reference-count} are placeholders.

The core SQL[12] commands summarized in Table 3 form the foundation for building SQL queries, often combined with operators (comparison, arithmetic, logical) to create more complex queries (since WHERE is used to filter data, it always works in conjunction with SQL operators). Aside from the basic SQL commands in Table 3 (Aggregating, Sorting, and Filtering), we define composite commands based on their combinations (i.e., Filtering+Aggregating and Filtering+Sorting). Using these, we manually create SQL query templates targeting different topics related to key elements of scientific articles, including Author Count, Author List, Reference Count, Title List, and Title Word Count. This process is illustrated in Figure 1 (right panel) and example templates are shown in Table 4; Note that some templates include placeholders to be instantiated with specific values. Finally, to capture authorship and citation relations, we introduce Relational Filtering and the topics of Author Relation and Citation Relation, which specifically target authorship and citation relations.

We collectively refer to the SQL commands in Table 4 as information processing skills, since they test different information processing capabilities. We have various templates per skill, designed to be applicable across all article collections. For each collection, we randomly select 10 templates and instantiate them with collection-specific values for all placeholders. We then execute these queries against the corresponding database to generate ground-truth answers.

### 3.3 Converting SQL Queries to Natural Language

We use Qwen2.5-Coder-32B-Instruct (Hui et al., 2024) to convert SQL queries into natural language questions. To ensure queries and questions are meaning preserving, we validate each generated question by converting it back to SQL and verifying that both queries produce identical results when executed against the collection database. For each SQL-collection pair, we repeat this process up to 10 times; if no valid question is obtained, we discard the query for that collection. Using the prompts in Appendix C, we successfully generate natural language questions for 82.9% of SQL-collection pairs. Following this, we obtain 2,121 test questions for evaluating models against the four context lengths defined in Section 3.1, and 19,543 instances for training (see Section 4.2). Table 5 presents descriptive statistics for the SciTrek test set. For each context length, our test partition covers all information processing skills and question topics described in Section 3.2. The distribution of SQL commands and operators is provided in Appendix D.

### 3.4 Data Quality Validation

To validate the quality of SciTrek, we used Prolific[13] to crow-source annotations by asking human annotators to provide answers to our natural language questions. We randomly sampled 120 instances representing the six skills listed in Table 4 (20 instances per skill). Each instance was

---

[12]Structured Query Language (SQL) is a standardized language for managing and querying relational databases.

[13]https://www.prolific.com/

| Length | Instances | Articles/C | Words/Q | Words/A |
|---|---|---|---|---|
| 64K | 112 | 4.2 | 14.6 | 6.2 |
| 128K | 728 | 5.7 | 16.9 | 7.9 |
| 512K | 667 | 22.7 | 18.1 | 30.9 |
| 1M | 614 | 46.3 | 18.6 | 63.6 |

Table 5: Descriptive statistics for SciTrek (test set). Length: the length level of input tokens; Instances: number of question-answer pairs with their contexts; Articles/C: average number of scientific articles per context; Words/Q: average number of words per question; Words/A: average number of words per answer.

| Skill | Agree (%) | Align (%) |
|---|---|---|
| Aggregating | 85.7 | 85.7 |
| Sorting | 85.0 | 80.0 |
| Filtering | 85.7 | 85.7 |
| Filtering+Aggregating | 95.0 | 85.0 |
| Filtering+Sorting | 89.5 | 89.5 |
| Relational Filtering | 89.5 | 73.7 |
| All | 88.3 | 83.3 |

Table 6: Inter-annotator agreement measured on a sample of 120 answers. Answers obtained from executing SQL queries are considered aligned if they match those provided by two or more annotators.

| Model | Context Size | Full-text Articles | | | | Database Tables | | | |
|---|---|---|---|---|---|---|---|---|---|
| | | F-64 | F-128 | F-512 | F-1024 | D-64 | D-128 | D-512 | D-1024 |
| Qwen2.5-7B-Instruct-1M | 1M | 4.5 | 2.8 | 0.3 | 0.0 | 20.5 | 14.3 | 5.0 | 2.0 |
| Qwen2.5-14B-Instruct-1M | 1M | 8.3 | 6.5 | 1.6 | 0.1 | 33.3 | 27.2 | 11.0 | 5.9 |
| Qwen3-4B-Instruct-2507 | 256K | 2.1 | 7.2 | — | — | 25.9 | 16.9 | 6.9 | 2.8 |
| Qwen3-4B-Thinking-2507 | 256K | 41.1 | 29.3 | — | — | 90.2 | 83.1 | 71.8 | 52.3 |
| Qwen3-30B-A3B-Instruct-2507 | 256K | 5.4 | 3.2 | — | — | 29.8 | 21.2 | 6.0 | 4.2 |
| Qwen3-30B-A3B-Thinking-2507 | 256K | 53.3 | 42.0 | — | — | 92.3 | 86.2 | 73.5 | 61.1 |
| Gemma-3-27B-IT | 128K | 6.2 | 3.4 | — | — | 31.8 | 25.0 | 11.3 | 6.1 |
| Llama-4-Scout-17Bx16E-Instruct | 10M | 5.4 | 2.8 | 1.3 | 1.1 | 28.6 | 19.0 | 7.4 | 4.0 |
| Llama-3.3-70B-Instruct | 128K | 8.3 | 3.5 | — | — | 47.0 | 36.2 | 14.6 | 8.1 |
| DeepSeek-R1-Distill-Llama-70B | 128K | 22.0 | 6.0 | — | — | 83.3 | 74.2 | 56.8 | 42.2 |
| Gemini 2.5 Pro | 1M | 41.7 | 26.0 | ⋆ | ⋆ | 91.7 | 83.5 | 55.4 | 31.5 |
| GPT-4.1 | 1M | 21.1 | 11.7 | 3.9 | 2.5 | 69.3 | 53.8 | 24.8 | 16.1 |
| o4-mini | 195K | 61.0 | 46.5 | — | — | 95.2 | 87.8 | 79.4 | 72.6 |

Table 7: Evaluation of open-weight (top) and proprietary (bottom) models on the SciTrek benchmark using exact match (%). Results are reported for two settings: using the full-text scientific articles as context and using the corresponding database tables. CLen denotes the maximum context length supported by each model. SciTrek comprises four article collections with token sizes of 64K, 128K, 512K, and 1M. — indicates the model cannot handle the given context size; ⋆ indicates that the model was not evaluated due to prohibitive computational cost. F-64/128/512/1024: test data with full-article inputs in different maximum lengths, 64K, 128K, 512K and 1M. D-64/128/512/1024: test data with underlying textual databases of F-64/128/512/1024 as inputs (on average about 2K tokens long).

independently annotated by three annotators. Annotators were asked to answer the questions using database tables rather than full-text articles, as documents spanning 1M tokens are impractical for humans to review. The metadata in the database tables were sourced from Semantic Scholar or generated through simple preprocessing as explained in Section 3.1, with manual corrections applied when necessary to make them aligned with the collected full-text articles in Markdown. More annotation details are in Appendix E. We measured inter-annotator agreement and the alignment of answers obtained from executing SQL queries on the database with human responses, using exact match. Annotators were considered in agreement if two or more provided the same answer to a question. Similarly, our answer was considered aligned with the humans if it matched the answer of two or more annotators. Table 6 reports the average agreement and alignment across 120 instances, showing that human annotators largely agree with each other and that our database-based answers are highly consistent with human responses. We also manually checked all 120 questions that were given to human annotators. Amongst the Relational Filtering in Table 6, we found 3 out of 19 questions ambiguous, leading to the gap between the alignment and agreement for the category. Although the conversion by Qwen2.5-Coder-32B-Instruct from SQL queries to natural language questions is fairly accurate, the complexity of SQL queries in Relational Filtering may introduce ambiguity in the questions.

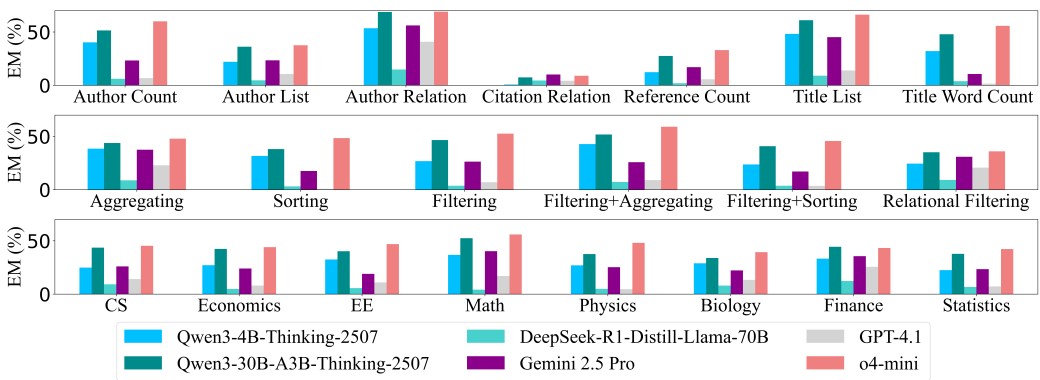

Figure 2: Fine-grained performances of Qwen3-4B-Thinking-2507, Qwen3-30B-A3B-Thinking-2507, DeepSeek-R1-Distill-Llama-70B, Gemini 2.5 Pro, GPT 4.1 and o4-mini in terms of exact match (EM) across question topics (top), information processing skills (middle) and subjects (bottom) when using full-text articles as context with an input length of 128K.

## 4 RESULTS

In our experiments, we evaluate both proprietary and open-weight models from various families that support contexts exceeding 128K and have shown strong performance on language understanding and mathematical reasoning benchmarks, e.g., MMLU (Hendrycks et al., 2021a) and MATH (Hendrycks et al., 2021b). For each family, we select the largest model we can feasibly run with our resources.[14] For example, we use Llama-4-Scout-17Bx16E-Instruct instead of Llama-4-Maverick-17B-128E-Instruct, and the distilled variant of DeepSeek-R1. Table 7 provides an overview of the models we consider, most of which were released within the past six months. For detailed model descriptions and settings, refer to Appendix G. As shown, the models vary in parameter scale and supported context length. We evaluate models in two context settings: (1) using the full-text scientific articles within a given collection as context and (2) using only the corresponding database tables. All models in our study are instruction-tuned, and we employ a uniform set of prompts across them, as detailed in Appendix F. Models generate three answers for each question. We assess performance using average exact match and F1, as the expected outputs are factual items with minimal variation, such as specific numbers, author names, or article titles.

### 4.1 ZERO-SHOT PROMPTING

Our zero-shot results are summarized in Table 7. Overall, we observe that SciTrek is challenging especially when using the full-text articles as contexts. In this setting, performance drops for all models as the input gets longer. This trend also manifests itself when using database tables as context. Perhaps unsurprisingly, proprietary models significantly outperform open-weight ones. We observe similar tendencies when using F1 as the evaluation metric (see Appendix H).

Figure 2 presents a fine-grained analysis for the best six models across question topics (e.g., author count, reference count, citation relations), skills (e.g., aggregating, sorting), and subjects (e.g., Economics, Biology) when using full-text articles as context. As can be seen, model performance shows little variation by subject. However, most models struggle more with sorting tasks, while performing better on aggregation. Performance is lowest on citation-related questions (i.e., Citation Relation, Reference Count), and somewhat higher on author- and title-related questions.

### 4.2 LONG-CONTEXT POST-TRAINING

Leveraging the data generation methodology described in Section 3, we curate a substantial training dataset (19,543 instances) across four context lengths (64K, 128K, 512K, and 1M). We use this data to assess whether more training could improve the performance of open-weight models on full-text articles. We experiment with two

---

[14]All our experiments were conducted on 4 NVIDIA HGX H200s.

well-established techniques: supervised fine-tuning (SFT) and reinforcement learning (RL). Because of the high computational cost, for this suite of experiments, we only report results with Qwen2.5-7B-Instruct-1M on data with maximum context of 128K tokens (7,703 training instances). Moreover, we examine whether the post-trained model can generalize along the dimensions of input length, question topics, and information processing skills. We compare in-distribution performance where the model is trained and tested in similar conditions (e.g., on lengths of 64K and 128K) to out-of distribution where the model is tested on an unseen dimension.

Specifically, we train the model on data corresponding to 64K and 128K context lengths and evaluate it both in-distribution (64K, 128K) and out-of-distribution (512K, 1M). For question topic, training uses author- and title-related questions at 64K and 128K, with evaluation covering the same topics in-distribution and citation-related questions out-of-distribution. For skills, training is performed on non-relational questions (Aggregating, Sorting, Filtering, Filtering+Sorting, Filtering+Aggregating), with evaluation including these in-distribution skills and the out-of-distribution Relational Filtering skill. For SFT, we train the model for 500 steps with a batch size of 32, a learning rate of $2 \times 10^{-6}$, and a warm-up rate of 0.05. For RL, we use GRPO (Shao et al., 2024) with a mixed reward of EM and F1 to encourage the model to produce both reasoning traces and answers. GRPO was chosen due to its success in similar verifiable and long-context tasks (Shao et al., 2024; Mroueh, 2025; Gurung & Lapata, 2025; Zheng et al., 2025). We found a simple sum of EM and F1 to improve accuracy over time while still providing a useful training signal from difficult questions. The prompt for generating reasoning traces is provided in Appendix I. Since GRPO optimization is time-consuming, we restrict training to a single epoch which takes around 5 days.

| Models | Length | | Topic | | Skills | |
|---|---|---|---|---|---|---|
| | ID | OOD | ID | OOD | ID | OOD |
| Qwen2.5 (ZS) | 3.1 | 0.2 | 3.9 | 1.5 | 1.4 | 5.8 |
| Qwen2.5 (SFT) | 16.3 | 2.3 | 20.9 | 10.0 | 10.8 | 19.2 |
| Qwen2.5 (GRPO) | 22.5 | 2.0 | 30.6 | 7.5 | 20.0 | 26.8 |
| GPT-4.1 (ZS) | 13.0 | 3.4 | 17.3 | 5.0 | 8.0 | 21.5 |
| Gemini 2.5 Pro (ZS) | 28.1 | — | 36.0 | 13.5 | 24.9 | 33.7 |
| o4-mini (ZS) | 48.4 | — | 62.5 | 22.3 | 53.7 | 39.3 |

Table 8: Post-training Qwen2.5-7B-Instruct-1M on SciTrek. Results with SFT and GRPO are compared with zero-shot prompting (ZS) of the same model and other proprietary models in terms of exact match (%). ID is a shorthand for in-distribution, and OOD for out-of-distribution. —: the model cannot handle the context size or has prohibitive computational cost.

The results in terms of exact match in Table 8 demonstrate that both SFT and GRPO slightly improve performance across dimensions, surpassing GPT-4.1 despite its much larger parameter size. However, Qwen2.5-7B-Instruct-1M is still not able to generalize to longer inputs, but does improve on questions related to out-of-distribution question topics and information processing skills.

## 5 MODEL FAILURE ANALYSES

We conduct additional analyses to understand model failures on SciTrek when using full-text articles as context. Table 9 shows correlations between model performance and various factors: the input article count, the question length (LenQ), the length of the underlying SQL query (LenSQL), and the answer length (LenA). They indicate that model performance correlates primarily with the question length, while o4-mini shows correlations with the input article count and the SQL query length.

Inspection of model answers reveals several failure patterns: (1) weaker models tend to respond by simply outputting "NULL", which suggests they rely more on the instruction to produce a fallback answer rather than genuinely understanding the given context; (2) models often fail to follow the specified output formats, e.g., returning author lists when author numbers are requested; (3) models sometimes provide incomplete answers when lists are requested for questions that require aggregation; and (4) models misinterpret compound conditions, particularly struggling with negation-based filtering (e.g., the last example in Table 1). Detailed analyses are provided in Appendix J.1.

We also manually analyzed 200 reasoning traces from Qwen2.5-7B-1M (GRPO), Qwen3-4B-A3B-Thinking-2507 and DeepSeek-R1-Distill-Llama-70B for 25 random SQL query templates (see more details in Appendix J.2). Specifically, we examined whether the reasoning is logically sound and individual steps obtain accurate information from specific contexts. We find that (1) the overall reasoning remains logically sound at 64K and 128K context lengths for all the three models, al-

| Model | #Articles | Question Length | SQL Length | Gold Answer Length |
|---|---|---|---|---|
| Qwen2.5-7B-1M | $-0.04$ | $\mathbf{-0.15}^*$ | 0.11 | $-0.01$ |
| Qwen2.5-14B-1M | $-0.04$ | $\mathbf{-0.16}^*$ | 0.07 | $-0.06$ |
| Qwen3-4B-Instruct-2507 | $-0.01$ | $\mathbf{-0.16}^*$ | 0.05 | $-0.03$ |
| Qwen3-4B-Thinking-2507 | $-0.06$ | $-0.14^*$ | $\mathbf{-0.19}^*$ | $-0.01$ |
| Qwen3-30B-A3B-Instruct-2507 | $-0.02$ | $\mathbf{-0.15}^*$ | 0.06 | 0.08 |
| Qwen3-30B-A3B-Think-2507 | $-0.12^*$ | $-0.08$ | $\mathbf{-0.19}^*$ | $-0.02$ |
| Gemma-3-27B-IT | $-0.08$ | $\mathbf{-0.14}^*$ | 0.02 | $-0.08$ |
| Llama-4-Scout | $-0.10^*$ | $\mathbf{-0.17}^*$ | $-0.03$ | $-0.03$ |
| Llama-3.3-70B | $-0.06$ | $\mathbf{-0.16}^*$ | $-0.03$ | $-0.05$ |
| DeepSeek-R1-Distill-Llama-70B | $-0.15^*$ | $\mathbf{-0.16}^*$ | $-0.02$ | $-0.08$ |
| Gemini 2.5 Pro | $-0.16^*$ | $\mathbf{-0.22}^*$ | $-0.03$ | $-0.01$ |
| GPT-4.1 | $-0.07$ | $\mathbf{-0.21}^*$ | 0.06 | 0.06 |
| o4-mini | $-0.14^*$ | $-0.09$ | $\mathbf{-0.25}^*$ | $-0.06$ |
| Qwen2.5-7B-Instruct-1M (SFT) | $-0.03$ | $\mathbf{-0.32}^*$ | 0.09 | $-0.14^*$ |
| Qwen2.5-7B-Instruct-1M (GRPO) | $-0.07$ | $\mathbf{-0.13}^*$ | $-0.02$ | $-0.08$ |

Table 9: Pearson correlation between various factors and model performance using exact match (with full-text articles as context, 128K token length). The lengths of questions, SQL queries, and gold answers are computed by counting words separated by spaces. Bold values indicate the strongest correlation for each model, and $^*$ denotes correlations with p-value $<0.05$. Zero-shot models are shown in the first block, supervised versions of Qwen2.5-7B-Instruct-1M are shown in the second block.

though the model occasionally includes repeated steps or extraneous operations unrelated to the question; and (2) despite not being trained for 512K, Qwen2.5-7B-1M (GRPO) still demonstrates coherent reasoning, but with reduced accuracy in specific steps compared to 64K and 128K; and (3) most reasoning steps that require counting are not accurate especially for references, which leads to incorrect answers from these models especially for Qwen2.5-7B-1M (GRPO). These findings indicate that GRPO-based reinforcement learning improves abstract reasoning, without enhancing the model's accuracy in fine-grained operations, such as counting the number of references.

## 6 CONCLUSION

This paper introduced SciTrek, a benchmark designed for testing the ability of LLMs to perform multi-document information synthesis and structured reasoning over full-text scientific articles. By generating questions and ground-truth answers through a SQL backbone over article metadata, we provide a framework with explicit reasoning processes that is highly scalable and enables fine-grained error analysis. Our extensive evaluation demonstrates that SciTrek poses a significant challenge to both open-weight and proprietary LLMs, with only modest performance gains observed through supervised fine-tuning and reinforcement learning. Specifically, models struggle with compound logical conditions, fail at tasks requiring sorting, and often produce incomplete or badly formatted outputs. We believe the proposed methodology for creating SciTrek generalizes beyond titles, authors, and references to encompass broader metadata elements and domains, offering a tool to diagnose persistent shortcomings in LCLMs and improve their capability to support scientific workflows.

## LIMITATIONS

While benchmarking fundamental reasoning capabilities of LCLMs over scientific articles, SciTrek only cover a limited set of core elements in scientific articles namely titles, authors, and references. We will explore more long-context capabilities in understanding scientific articles in future work, e.g., interpreting figures and domain-specific content reasoning.

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

## A EXAMPLE EXPERT-WRITTEN QUESTIONS

Our focused model capabilities are based on atomic SQL commands (i.e., sorting, filtering, and aggregating) and operators (i.e., comparison, arithmetic and logical), shown in Tables 3 and 4. These targeted reasoning capabilities are fundamental to complex tasks. QMDR (Wolfson et al., 2020) presents a widely used formalism, in which complex questions can be decomposed into atomic, simple questions that require filtering, aggregating, comparing, sorting, logical and arithmetic operations. A recent work, MoNaCo (Wolfson et al., 2025), further shows that real-world complex questions on multi-document understanding often demand aggregating and arithmetic skills based on their manual decomposition of naturally occurring human-written questions.

We build a benchmark on these capabilities as they are explicitly required in realistic scientific tasks. For example, expert-written literature review questions from OpenScholar (Asai et al., 2024) implicitly require these capabilities. There are five example questions from their released data and corresponding capabilities they require.

- Example Question 1: "What are the latest works on finetuning an auto-regressive LM for dense passage retrieval? How are their performance compared with bi-directional encoders?" (Required Fundamental Capabilities: sorting, arithmetic, logical)
- Example Question 2: "Which downstream task can solved by AlphaFold3 but cannot performed by ESM-3?" (Required Fundamental Capabilities: filtering, logical)
- Example Question 3: "Citation graph is a good tool to find relevant work and to help understand the evolution of the domain. What recent research has been done to analyze scientific discourse and generate citation content using citation networks and related textual information?" (Required Fundamental Capabilities: sorting)
- Example Question 4: "What types of mechanical resonators have been used to couple with superconducting qubits?" (Required Fundamental Capabilities: filtering)
- Example Question 5: "Compared to 2023, how has the percentage of finished goods apparel factories from countries other than Vietnam, China, and Cambodia changed in 2024?" (Required Fundamental Capabilities: arithmetic, aggregating, comparing)

These capabilities are critical to real-world tasks, even in expert-written example questions in the financial domain from LongBench v2 (Bai et al., 2025) (shown in the example question below).

- Example Question: "In the financial reports of Apple Inc. and Samsung Electronics for the years 2022 and 2023, which company has a higher percentage of revenue derived from the product category of phones, and in what range do the differences in this dependency between the two companies in the two years fall?" (Required Fundamental Capabilities: comparing, filtering, aggregating, logical, and arithmetic)

While OpenScholar and LongBench v2 have more realistic questions, they do not support detailed error analysis to check where models fail as there are no labels of the required capabilities and they are not scalable because of heavy human effort in their data construction. To proxy this, we build our scalable benchmark with questions and answers approximated by database simulation, and we provide an explicit label of reasoning capabilities for each question.

## B DESCRIPTION OF SCITREK'S DATABASE

We construct a database with three tables for each article collection, using metadata from Semantic Scholar along with basic preprocessing. A description of these tables is provided in Table 10.

## C PROMPTS FOR CONVERTING BETWEEN SQL QUERIES AND NATURAL LANGUAGE QUESTIONS

We generate natural language questions by prompting Qwen2.5-Coder-32B to convert SQL queries to natural language. Our prompt is presented in Figure 3. The prompt that we use to convert natural language questions back to to SQL queries is shown in Figure 4.

| Database Table | Column Name | Data Type | Description |
|---|---|---|---|
| *articles* | article_id | String | the unique identifier of the article |
| | article_title | String | the title of the article |
| | title_word_count | Integer | the number of words in the article's title (using spaces to determine word boundaries) |
| | author_count | Integer | the number of authors in the article |
| | reference_count | Integer | the number of references that are cited in the article |
| *article-author* | relation_id | String | the unique identifier of the article-author relations |
| | article_id | String | the identifier of the associated article |
| | author_name | String | the name of the author/s |
| | author_position | Integer | the position of the author in the author list (starting from 0 for the first author) |
| *citing-cited* | relation_id | String | the unique identifier of the citation relations between two articles |
| | article_id_citing | String | the identifier of the article that cites another article |
| | article_id_cited | String | the identifier of the article that is cited by another article |

Table 10: Description of the database tables that we use to curate SQL queries and answers.

## D  TEST SET COVERAGE OF SQL COMMANDS AND OPERATORS

Our test dataset covers all SQL commands and operators that are listed in Table 3. The detailed distribution is shown in Table 11.

| Command/Operator | #Instances | Proportion (%) |
|---|---|---|
| SELECT | 2,121 | 100.00% |
| WHERE | 1,821 | 85.86% |
| = | 1,032 | 48.66% |
| IN | 682 | 32.15% |
| OR | 616 | 29.04% |
| ORDER BY | 591 | 27.86% |
| < | 476 | 22.44% |
| > | 450 | 21.22% |
| COUNT | 370 | 17.44% |
| ASC | 301 | 14.19% |
| DESC | 287 | 14.19% |
| DISTINCT | 280 | 13.20% |
| * | 215 | 10.14% |
| MAX | 213 | 10.04% |
| AND | 195 | 9.19% |
| GROUP BY | 175 | 8.25% |
| <= | 175 | 8.25% |
| % | 164 | 7.73% |
| >= | 154 | 7.26% |
| NOT | 154 | 7.26% |
| <> | 153 | 7.21% |
| AVG | 135 | 6.36% |
| MIN | 133 | 6.27% |
| BETWEEN | 106 | 5.00% |
| SUM | 102 | 4.81% |
| / | 100 | 4.71% |
| LIKE | 50 | 2.36% |
| + | 48 | 2.26% |
| − | 23 | 1.08% |

Table 11: Distribution of SQL commands and operators covered in our test data.

---

**Instruction to Convert SQL Queries to Natural Language Questions**

You are given a database with three tables: *articles*, *article-author*, and *citing-cited*.

The *articles* table contains the following columns:

- article_id (String): the unique identifier of the article;
- article_title (String): the title of the article;
- title_word_count (Integer): the number of words in the article's title (using spaces to determine word boundaries);
- author_count (Integer): the number of authors for the article;
- reference_count (Integer): the number of references cited in the article.

The *article-author* table contains the following columns:

- relation_id (String): the unique identifier of the article-author relationship;
- article_id (String): the identifier of the associated article;
- author_name (String): the name of the author;
- author_position (Integer): the position of the author in the author list (starting from 0 for the first author).

The *citing-cited* table contains the following columns:

- relation_id (String): the unique identifier of the citation relationship between two articles;
- article_id_citing (String): the identifier of the article which cites the other article;
- article_id_cited (String): the identifier of the article which is cited by the other article.

Assumptions:

- The *articles* table contains multiple entries;
- The *article_author* table maps authors to articles, where one author can contribute to multiple articles, and one article can have multiple authors;
- The *citing_cited* table represents citation relationships among articles in the articles table, where one article can be cited by multiple others.

Your task involves two steps:

1. Understand the given SQL query in the context of the database schema described above;
2. Convert the SQL query into a clear and natural-sounding question in everyday language, as if you were reading textural articles rather than querying a database.

The given SQL query: {sql_query}

Do not refer to relation_id or article_id in the natural-language question.
You must output the SQL query and the corresponding question in the following JSON format, and do not include any extra text:

{"sql": "the given SQL query", "question": "the generated question"}

---

Figure 3: Prompt template for converting SQL queries to natural language questions.

## E  HUMAN ANNOTATION DETAILS

We validated the quality of our curated questions and answers with a human annotation study. We recruited crowdworkers via Prolific (https://www.prolific.com/), specifically native English speakers from the US or UK. Our annotators were compensated above the UK living wage, at £12 per hour. We randomly sampled 120 instances from our test dataset (20 per skill in Table 4). Each instance was annotated by three participants who were given the same instructions as those used for model testing with database tables as context (Figure 6, Appendix F). In each annotation session, crowdworkers were given three questions with different contexts. To ensure annotation quality, each session included a quality control question. On average, annotators spent about 5.5 minutes per session for the three questions.

---

**Instruction to Convert Natural Language Questions to SQL Queries**

You are given a database with three tables: *articles*, *article-author*, and *citing-cited*.

The *articles* table contains the following columns:

- article_id (String): the unique identifier of the article;
- article_title (String): the title of the article;
- title_word_count (Integer): the number of words in the article's title (using spaces to determine word boundaries);
- author_count (Integer): the number of authors for the article;
- reference_count (Integer): the number of references cited in the article.

The *article-author* table contains the following columns:

- relation_id (String): the unique identifier of the article-author relationship;
- article_id (String): the identifier of the associated article;
- author_name (String): the name of the author;
- author_position (Integer): the position of the author in the author list (starting from 0 for the first author).

The *citing-cited* table contains the following columns:

- relation_id (String): the unique identifier of the citation relationship between two articles;
- article_id_citing (String): the identifier of the article which cites the other article;
- article_id_cited (String): the identifier of the article which is cited by the other article.

Assumptions:

- The *articles* table contains multiple entries;
- The *article_author* table maps authors to articles, where one author can contribute to multiple articles, and one article can have multiple authors;
- The *citing_cited* table represents citation relationships among articles in the articles table, where one article can be cited by multiple others.

Available core SQL commands:

- Aggregating: `MIN()`, `MAX()`, `COUNT()`, `SUM()`, `AVG()`, `DISTINCT`
- Filtering: `WHERE`
- Organizing: `ORDER BY`, `ASC`, `DESC`, `GROUP BY`

Available core SQL operators:

- Comparison: $=, >, <, >=, <=, <>$, `LIKE`
- Arithmetic: $+, -, *, /, \%$
- Logical: `AND`, `NOT`, `OR`, `BETWEEN`, `IN`

Your task is to:

1. Understand the database schema described above and the given natural language question below;
2. Convert the natural language question into a SQL query in the context of the database schema with the listed SQL commands and operators.

The given natural language question: {question}

Do not output relation_id or article_id in generated SQL query.
Use the SQL commands and operators listed above.
Make the generated SQL query aligned well with the natural language question.
You must output the natural language question and the generated SQL query in the following JSON format, and do not include any extra text:
{"question": "the given question", "sql": "the generated SQL query"}

---

Figure 4: Prompt template for converting natural language questions to SQL queries.

We also manually checked all 120 questions that were given to human annotators. Amongst the Relational Filtering in Table 6, we found 3 out of 19 questions ambiguous, leading to the gap between the alignment and agreement for the category. We used Qwen2.5-Coder-32B-Instruct to

convert SQL queries to natural language questions. Although this mapping is fairly accurate, the complexity of SQL queries in Relational Filtering may introduce ambiguity in the questions.

## F  PROMPTS FOR MODEL TESTING

We employ two different prompts for (zero-shot) model evaluation. The prompt for assessing LCLM capabilities against full-text articles is presented in Figure 5, and the prompt for using the corresponding database tables as the context is shown in Figure 6.

---

**Testing Instruction Using Full-text Articles**

Articles:

{scientific articles}

You are provided with multiple scientific articles above. Based on the information in these articles, answer the question provided below.

If the answer consists of multiple components (e.g., author names, article titles, reference counts), separate them with commas.
For example, if the answer includes two author names, your response should be in the format of 'the-first-author-name, the-second-author-name'.
When counting the number of words in article titles, use spaces to determine word boundaries. Words are spaced apart individually.

Respond with only the final answer, with no additional explanation or formatting. If you cannot get the answer from the articles, just return 'NULL'.

Question:

{question}

---

Figure 5: Prompt template using full-text articles as context.

## G  EXPERIMENTAL DETAILS

The details of the models we use in our experiments are presented in Table 12. For all models, we use their default inference configuration for answer generation from their Huggingface repositories.

We include both reasoning models and non-reasoning models. Qwen3-4B-Thinking-2507, Qwen3-30B-A3B-Thinking-2507, DeepSeek-R1-Distill-Llama-70B, Gemini 2.5 Pro and o4-mini are reasoning models. In our experiments, o4-mini conducts reasoning with the default thinking effort *medium*, Gemini 2.5 Pro was given a thinking budget of 512 tokens, and Qwen3-4B-Thinking-2507, Qwen3-30B-A3B-Thinking-2507, and DeepSeek-R1-Distill-Llama-70B were prompted with the reasoning prompt in Figure 7.

## H  ADDITIONAL RESULTS

Table 13 presents additional results using F1 as the evaluation metric. Models are tested against full-text scientific articles and database tables as context. The input length averages only 1,980 tokens when database tables of are used as context across different lengths of full-text article collections.

## I  REASONING TRACES FOR REINFORCEMENT LEARNING

The prompt used by GRPO to generate reasoning traces is given in Figure 7.

---

**Testing Instruction Using Database Tables**

You are given three tables, named *articles*, *article-author*, and *citing-cited*.

The *articles* table contains the following columns:

- article_id (String): the unique identifier of the article;
- article_title (String): the title of the article;
- title_word_count (Integer): the number of words in the article's title (using spaces to determine word boundaries);
- author_count (Integer): the number of authors for the article;
- reference_count (Integer): the number of references cited in the article.

{the table of articles}

The *article-author* table contains the following columns:

- relation_id (String): the unique identifier of the article-author relationship;
- article_id (String): the identifier of the associated article;
- author_name (String): the name of the author;
- author_position (Integer): the position of the author in the author list (starting from 0 for the first author).

{the table of article-author}

The *citing-cited* table contains the following columns:

- relation_id (String): the unique identifier of the citation relationship between two articles;
- article_id_citing (String): the identifier of the article which cites the other article;
- article_id_cited (String): the identifier of the article which is cited by the other article.

{the table of citing-cited}

Based on the information in the tables above, answer the question provided below.

If the answer consists of multiple components (e.g., author names, article titles, reference counts), separate them with commas. For example, if the answer includes two author names, your response should be in the format of 'the-first-author-name, the-second-author-name'.

Respond with only the final answer, with no additional explanation or formatting. If you cannot get the answer from the tables, just return 'NULL'.

Question: {question}

---

Figure 6: Prompt template using database tables as context.

## J   DETAILS FOR MODEL FAILURE ANALYSES

### J.1   ANALYSES ON MODEL ANSWERS

To gain deeper insights into model behavior, we analyzed the failure patterns exhibited in model-generated answers on SciTrek. We randomly selected 60 instances in the input length of 128K that were hard for all zero-shot and supervised models (10 instances per skill in Table 4). We find that models tend to output "NULL" particularly on samples that require the skill of filtering. As shown in Table 14, most open-weight models tend to randomly output "NULL" for any sample. This indicates that these models rely more on the instruction to produce a fallback "NULL" answer rather than on genuine understanding of the provided context.[15] In addition, we find that models fail to follow the specified format for the answers, especially for sorting-related questions, as shown in Table 15. For

---

[15]Our testing instruction prompted models to generate "NULL" when they could not get the answer from the context.

| Model | #Parameters | Open Weights | Context Length | Release Date |
|---|---|---|---|---|
| Qwen2.5-7B-Instruct-1M | 7B | ✓ | 1,010,000 | Jan 2025 |
| Qwen2.5-14B-Instruct-1M | 14B | ✓ | 1,010,000 | Jan 2025 |
| Qwen3-4B-Instruct-2507 | 4B | ✓ | 262,144 | July 2025 |
| Qwen3-4B-Thinking-2507 | 4B | ✓ | 262,144 | July 2025 |
| Qwen3-30B-A3B-Instruct-2507 | 3Bx10E | ✓ | 262,144 | July 2025 |
| Qwen3-30B-A3B-Thinking-2507 | 3Bx10E | ✓ | 262,144 | July 2025 |
| Gemma-3-27B-IT | 27B | ✓ | 131,072 | Mar 2025 |
| Llama-4-Scout-17Bx16E-Instruct | 17Bx16E | ✓ | 10,485,760 | Apr 2025 |
| Llama-3.3-70B-Instruct | 70B | ✓ | 131,072 | Dec 2024 |
| DeepSeek-R1-Distill-Llama-70B | 70B | ✓ | 131,072 | Jan 2025 |
| Gemini 2.5 Pro | — | ✗ | 1,048,576 | Jun 2025 |
| GPT-4.1 | — | ✗ | 1,047,576 | Apr 2025 |
| o4-mini | — | ✗ | 200,000 | Apr 2025 |

Table 12: Proprietary and open-weight LCLMs that are evaluated in our experiments. (Parameter numbers for proprietary models are not accessible. "17B×16E" indicates that the model uses a mixture-of-experts architecture with 16 experts, each containing 17 billion parameters. Context length is based on the maximum number of tokens these models can handle as input.)

| Model | Context Size | Full-text Articles | | | | Database Tables | | | |
|---|---|---|---|---|---|---|---|---|---|
| | | F-64 | F-128 | F-512 | F-1024 | D-64 | D-128 | D-512 | D-1024 |
| Qwen2.5-7B-Instruct-1M | 1M | 8.7 | 7.0 | 1.1 | 0.0 | 39.2 | 39.3 | 28.7 | 21.2 |
| Qwen2.5-14B-Instruct-1M | 1M | 20.6 | 17.5 | 7.4 | 1.0 | 58.2 | 53.9 | 44.8 | 34.8 |
| Qwen3-4B-Instruct-2507 | 256K | 7.2 | 7.3 | — | — | 47.2 | 46.0 | 38.9 | 31.1 |
| Qwen3-4B-Thinking-2507 | 256K | 50.2 | 42.3 | — | — | 92.7 | 85.8 | 79.4 | 63.6 |
| Qwen3-30B-A3B-Instruct-2507 | 256K | 15.0 | 16.4 | — | — | 55.6 | 53.8 | 44.7 | 42.0 |
| Qwen3-30B-A3B-Thinking-2507 | 256K | 62.5 | 55.1 | — | — | 94.4 | 88.3 | 80.8 | 74.5 |
| Gemma-3-27B-IT | 128K | 22.6 | 14.7 | — | — | 52.8 | 52.5 | 47.8 | 40.7 |
| Llama-4-Scout-Instruct | 10M | 20.3 | 17.4 | 15.6 | 14.8 | 49.8 | 44.9 | 40.5 | 37.4 |
| Llama-3.3-70B-Instruct | 128K | 25.3 | 15.1 | — | — | 65.8 | 63.5 | 53.7 | 49.5 |
| DeepSeek-R1-Distill-Llama-70B | 128K | 29.0 | 9.0 | — | — | 85.5 | 77.6 | 68.0 | 62.3 |
| Gemini 2.5 Pro | 1M | 58.1 | 48.8 | ⋆ | ⋆ | 95.3 | 88.3 | 79.0 | 69.8 |
| GPT-4.1 | 1M | 36.0 | 29.7 | 22.3 | 19.6 | 82.8 | 73.3 | 61.8 | 56.3 |
| o4-mini | 195K | 74.0 | 63.8 | — | — | 97.2 | 89.2 | 87.0 | 86.5 |

Table 13: Evaluation of open-weight (top) and proprietary (bottom) models on the SciTrek benchmark using F1 (%). Results are reported for two settings: using the full-text articles as context and using the corresponding database tables. Context Size denotes the maximum context length supported by each model. SciTrek comprises four article collections with token sizes of 64K, 128K, 512K, and 1M. — indicates the model cannot handle the given context size; ⋆ indicates models not evaluated due to prohibitive computational cost. F-64/128/512/1024: test data with full-article inputs in different maximum lengths, 64K, 128K, 512K and 1M. D-64/128/512/1024: test data with underlying textual databases of F-64/128/512/1024 as inputs (on average about 2K tokens long).

example, models tend to generate author lists when sorted author numbers are requested and output lists when aggregates are requested. Interestingly, GPT-4.1 produces incorrectly formatted answers far more frequently than all other models. We further observe that some models tend to produce partial answers when lists are requested, especially for tasks that require aggregation. As shown in Table 16, our GRPO-based model tends to generate partial answers more than other models.

We also analyzed model performance on questions involving negation (e.g., those containing adverbs such as "not" or "never"). For questions that involve filtering-related skills, we have 132 instances with negation in our test set (Filtering: 11, Filtering+Aggregation: 15, Filtering+Sorting:12, and Relational Filtering:94). The results in Table 17 show that *all* models struggle with negation-based filtering.

## J.2 EXAMPLE REASONING TRACES FROM GRPO-BASED MODEL

> **Instruction Template to Generate Reasoning Traces**
>
> Articles:
>
> {scientific articles}
>
> You are provided with multiple scientific articles above. Based on the information in these articles, answer the question provided below.
>
> If the answer consists of multiple components (e.g., author names, article titles, reference counts), separate them with commas.
> For example, if the answer includes two author names, your response should be in the format of 'the-first-author-name, the-second-author-name'.
> When counting the number of words in article titles, use spaces to determine word boundaries. Words are spaced apart individually.
>
> Think step by step, and place your final answer within \boxed{}. If you cannot get the answer from the articles, just return 'NULL'.
>
> Question: {question}

Figure 7: Prompt template to generate reasoning traces in reinforcement learning using full-text articles as context.

| Model | Aggregation | Sorting | Filtering | Filtering +Aggregation | Filtering +Sorting | Relational Filtering |
|---|---|---|---|---|---|---|
| Qwen2.5-7B-Instruct-1M | 70 | 90 | 80 | 90 | 80 | 80 |
| Qwen2.5-14B-Instruct-1M | 10 | 10 | 20 | 50 | 50 | 60 |
| Gemma-3-27B-IT | 10 | 0 | 0 | 40 | 10 | 40 |
| Llama-4-Scout-14Bx16E-Instruct | 30 | 20 | 30 | 60 | 20 | 60 |
| Llama-3.3-70B-Instruct | 10 | 10 | 30 | 80 | 50 | 50 |
| DeepSeek-R1-Distill-Llama-70B | 0 | 0 | 0 | 10 | 0 | 10 |
| Gemini 2.5 Pro | 0 | 0 | 0 | 20 | 30 | 10 |
| GPT-4.1 | 0 | 0 | 0 | 10 | 0 | 0 |
| o4-mini | 0 | 0 | 10 | 10 | 30 | 20 |
| Qwen2.5-7B-Instruct-1M (SFT) | 0 | 0 | 40 | 30 | 40 | 0 |
| Qwen2.5-7B-Instruct-1M (GRPO) | 0 | 0 | 0 | 0 | 0 | 0 |

Table 14: Proportion of samples (%) where models output "NULL" broken down across information processing skills. Zero-shot models are shown in the first block, supervised versions of Qwen2.5-7B-Instruct-1M are shown in the second block.

| Model | Aggregation | Sorting | Filtering | Filtering +Aggregation | Filtering +Sorting | Relational Filtering |
|---|---|---|---|---|---|---|
| Qwen2.5-7B-Instruct-1M | 0 | 0 | 0 | 0 | 0 | 0 |
| Qwen2.5-14B-Instruct-1M | 0 | 30 | 10 | 0 | 30 | 0 |
| Gemma-3-27B-IT | 0 | 20 | 30 | 20 | 20 | 0 |
| Llama-4-Scout-14Bx16E-Instruct | 10 | 10 | 0 | 10 | 30 | 10 |
| Llama-3.3-70B-Instruct | 10 | 10 | 30 | 0 | 10 | 0 |
| DeepSeek-R1-Distill-Llama-70B | 0 | 0 | 0 | 0 | 0 | 0 |
| Gemini 2.5 Pro | 0 | 10 | 0 | 0 | 0 | 0 |
| GPT-4.1 | 30 | 30 | 20 | 50 | 20 | 0 |
| o4-mini | 10 | 20 | 0 | 0 | 0 | 0 |
| Qwen2.5-7B-Instruct-1M (SFT) | 0 | 0 | 0 | 0 | 0 | 0 |
| Qwen2.5-7B-Instruct-1M (GRPO) | 0 | 0 | 0 | 10 | 0 | 0 |

Table 15: Proportion of samples (%) where models do not follow the specified answer format broken down across information processing skills. Zero-shot models are shown in the first block, supervised versions of Qwen2.5-7B-Instruct-1M are shown in the second block.

We manually analyzed 200 reasoning traces from reasoning models including Qwen2.5-7B-1M (GRPO), Qwen3-4B-A3B-Thinking-2507 and DeepSeek-R1-Distill-Llama-70B for 25 random SQL

| Model | Aggregation | Sorting | Filtering | Filtering +Aggregation | Filtering +Sorting | Relational Filtering |
|---|---|---|---|---|---|---|
| Qwen2.5-7B-Instruct-1M | 10 | 0 | 10 | 0 | 0 | 0 |
| Qwen2.5-14B-Instruct-1M | 0 | 0 | 0 | 0 | 0 | 10 |
| Gemma-3-27B-IT | 10 | 10 | 10 | 0 | 0 | 0 |
| Llama-4-Scout-14Bx16E-Instruct | 0 | 0 | 0 | 0 | 0 | 0 |
| Llama-3.3-70B-Instruct | 0 | 0 | 10 | 0 | 0 | 0 |
| DeepSeek-R1-Distill-Llama-70B | 0 | 0 | 0 | 20 | 0 | 0 |
| Gemini 2.5 Pro | 20 | 0 | 10 | 20 | 0 | 10 |
| GPT-4.1 | 10 | 0 | 0 | 0 | 0 | 0 |
| o4-mini | 10 | 0 | 20 | 30 | 10 | 0 |
| Qwen2.5-7B-Instruct-1M (SFT) | 10 | 0 | 0 | 0 | 0 | 20 |
| Qwen2.5-7B-Instruct-1M (GRPO) | 20 | 0 | 30 | 10 | 20 | 0 |

Table 16: Proportion of samples (%) where models generate partial answers broken down across information processing skills. Zero-shot models are shown in the first block, supervised versions of Qwen2.5-7B-Instruct-1M are shown in the second block.

| Model | Filtering +Aggregation | Filtering +Sorting | Filtering | Relational Filtering | Average |
|---|---|---|---|---|---|
| Qwen2.5-7B-Instruct-1M | 0 | 0 | 0 | 0 | 0 |
| Qwen2.5-14B-Instruct-1M | 0 | 20.0 | 0 | 2.1 | 3.8 |
| Gemma-3-27B-IT | 0 | 6.7 | 0 | 0 | 0.8 |
| Llama-4-Scout-14Bx16E-Instruct | 0 | 20.0 | 0 | 2.1 | 3.8 |
| Llama-3.3-70B-Instruct | 0 | 6.7 | 0 | 0 | 0.8 |
| DeepSeek-R1-Distill-Llama-70B | 0 | 13.3 | 0 | 0 | 1.5 |
| Gemini 2.5 Pro | 0 | 20.0 | 8.3 | 6.4 | 9.1 |
| GPT-4.1 | 0 | 13.3 | 0 | 2.1 | 3.8 |
| o4-mini | 0 | 73.3 | 41.7 | 7.4 | 22.0 |
| Qwen2.5-7B-Instruct-1M (SFT) | 0 | 6.7 | 0 | 11.7 | 9.1 |
| Qwen2.5-7B-Instruct-1M (GRPO) | 0 | 33.3 | 8.3 | 3.2 | 8.3 |

Table 17: Model performance on questions that involve negation (e.g., "not" or "never"), using exact match (%). Zero-shot models are shown in the first block, supervised versions of Qwen2.5-7B-Instruct-1M are shown in the second block.

query templates. We cannot obtain reasoning traces from Gemini 2.5 Pro, GPT-4.1 and o4-mini for analysis. Table 18 shows example reasoning traces for three representative SQL query templates from our GRPO-based Qwen2.5-7B-Instruct-1M, Qwen3-4B- Thinking-2507 and DeepSeek-R1-Distill-Llama-70B. For each SQL query template, we have different SQL queries and questions for different sizes of contexts, while the questions require the same reasoning capability. There are no reasoning traces when the input is 512K or 1M tokens for Qwen3-4B-Thinking-2507 and DeepSeek-R1-Distill-Llama-70B, as they do not support the input of 512K or 1M tokens.

| Template 1 | `SELECT MIN(author_count) FROM articles WHERE reference_count > {reference-count}` |
|---|---|
| Question Context 64K | What is the smallest number of authors for articles that have more than 13 references? |
| Gold Answer | 2 |
| GRPO Reasoning | To solve this problem, we need to follow these steps: 1. Identify articles with more than 13 references. 2. Determine the number of authors in each of those articles. 3. Find the smallest number of authors among these articles. Let's go through each step in detail: ### Step 1: Identify articles with more than 13 references From the provided articles, we need to count the references in each article: 1. **"Best Constants in Young's Inequality, Its Converse, and Its Generalization to More than Three Functions"** - References: 13 (listed in the reference section) 2. **"Optical detection of strain and doping inhomogenieties in single layer MoS2"** - References: 62 (listed in the reference section) 3. **"Ambipolar device simulation based on the drift-diffusion model in ion-gated transition metal dichalcogenide transistors"** - References: 65 (listed in the reference section) 4. **"The Impact of AI in Physics Education: A Comprehensive Review from GCSE to University Levels"** - References: 32 (listed in the reference section) Only the articles titled "Optical detection of strain and doping inhomogenieties in single layer MoS2", "Ambipolar device simulation based on the drift-diffusion model in ion-gated transition metal dichalcogenide transistors", and "The Impact of AI in Physics Education: A Comprehensive Review from GCSE to University Levels" have more than 13 references. ### Step 2: Determine the number of authors in each of these articles Now, let's count the authors in each of these articles: 1. **"Optical detection of strain and doping inhomogenieties in single layer MoS2"** - Authors: A. Michail, N. Delikoukos, J. Parthenios, C. Galiotis, K. Papagelis (5 authors) 2. **"Ambipolar device simulation based on the drift-diffusion model in ion-gated transition metal dichalcogenide transistors"** - Authors: Akiko Ueda, Yijin Zhang, Nobuyuki Sano, Hiroshi Imamura, Yoshihiro Iwasa (5 authors) 3. **"The Impact of AI in Physics Education: A Comprehensive Review from GCSE to University Levels"** - Authors: Will Yeadon, Tom Hardy (2 authors) ### Step 3: Find the smallest number of authors Among the articles with more than 13 references, the smallest number of authors is 2. Thus, the smallest number of authors for articles that have more than 13 references is \boxed{2}. |

Table 18: Example reasoning traces for different SQL query templates and their corresponding questions from different models (GRPO-based Qwen2.5-7B-Instruct-1M, Qwen3-4B-Thinking-2507 and DeepSeek-R1-Distill-Llama-70B). (continued on the next page)

| Qwen3 Reasoning | I need to find the smallest number of authors for articles that have more than 13 references. |
|---|---|
| | Let me go through each article and count the number of references (citations) for each article. |
| | First, I'll look for the reference count in each article. |
| | Article 1 (The Best Constants in Young's Inequality...): This appears to be a paper with 30 references listed (from the references section at the end). The author count is 2 (HERM JAN BRASCAMP and ELLIOTT H. LIEB). |
| | Article 2 (Optical detection of strain and doping inhomogenieties in single layer MoS2): This paper has a list of authors at the top (A. Michail, N. Delikoukos, J. Parthenios, C. Galiotis, and K. Papagelis) - so 5 authors. The references section shows 33 references (from reference numbers 1-33). |
| | `<...we cut it down to save space...>` |
| | So the smallest number is 2. |
| | Actually, let me check if there are any other articles with more than 13 references that I might have missed. The problem states "You are provided with multiple scientific articles above", and I see 4 articles. |
| | Let me check the reference counts again: |
| | Article 1: "References" section has 30 entries (from [1] to [30]) Article 2: "References" section has 33 entries (from [1] to [33]) Article 3: "References" section has 65 entries (from [1] to [65]) Article 4: "References" section has 35 entries (from [1] to [35]) |
| | All have more than 13 references. |
| | The number of authors: Article 1: 2 authors Article 2: 5 authors Article 3: 5 authors Article 4: 2 authors |
| | So the smallest number of authors is 2. |
| | I think that's the answer. |
| | Actually, wait - I need to be careful with the reference counts. Let me count the references for each article. |
| | Article 1 (The Best Constants in Young's Inequality...): The references section has 30 references (from [1] to [30]) - so more than 13, 2 authors. |
| | `<...we cut it down to save space...>` |
| | So the smallest number of authors for articles with more than 13 references is 2. |
| | I think the answer is 2. |
| | Let me double check the author counts: |
| | Article 1: "HERM JAN BRASCAMP Department of Physics, Princeton University... AND ELLIOTT H. LIEB Departments of Mathematics and Physics, Princeton University" - 2 authors. |
| | `<...we cut it down to save space...>` |
| | To determine the smallest number of authors for articles that have more than 13 references, I need to examine each article's reference count and author count. |
| | Let's analyze each article: |
| | **Article 1: "The Best Constants in Young's Inequality..."** - Reference count: 30 references (from [1] to [30]) - Author count: 2 authors (HERM JAN BRASCAMP and ELLIOTT H. LIEB) - Has more than 13 references |
| | `<...we cut it down to save space...>` |
| | All four articles have more than 13 references. The author counts are: - Article 1: 2 authors - Article 2: 5 authors - Article 3: 5 authors - Article 4: 2 authors |
| | The smallest number of authors for articles with more than 13 references is 2. |
| | `\boxed{2}` |

Table 18: Example reasoning traces for different SQL query templates and their corresponding questions from different models (GRPO-based Qwen2.5-7B-Instruct-1M, Qwen3-4B-Thinking-2507 and DeepSeek-R1-Distill-Llama-70B). (continued on the next page)

| DeepSeek-R1 Reasoning | Alright, so I need to find the smallest number of authors for articles that have more than 13 references. Let me go through each article one by one and check the number of authors and their references.

Starting with the first article: It's authored by Will Yeadon and Tom Hardy. That's 2 authors. The references section shows 32 entries, which is way more than 13. So this article qualifies, and it has 2 authors.

Next, the second article has 5 authors: A. Michail, N. Delikoukos, J. Parthenios, C. Galiotis, and K. Papagelis. The references here are 31, which is more than 13. So this article also qualifies with 5 authors.

The third article is by Akiko Ueda, Yijin Zhang, Nobuyuki Sano, Hiroshi Imamura, and Yoshihiro Iwasa. That's 5 authors. The references count is 65, well above 13. So 5 authors here.

The fourth article lists two authors again: Will Yeadon and Tom Hardy. The references are 32, which is more than 13. So another instance with 2 authors.

Looking at these, the smallest number of authors among the qualifying articles is 2. Both the first and fourth articles have 2 authors and more than 13 references. Therefore, the smallest number is 2. </think>
The smallest number of authors for articles with more than 13 references is 2.
\boxed{2} |
|---|---|

| Template 1 | `SELECT MIN(author_count) FROM articles WHERE reference_count > {reference-count}` |
|---|---|
| Question Context 128K | What is the smallest number of authors for articles that have more than 25 references? |
| Gold Answer | 2 |

Table 18: Example reasoning traces for different SQL query templates and their corresponding questions from different models (GRPO-based Qwen2.5-7B-Instruct-1M, Qwen3-4B-Thinking-2507 and DeepSeek-R1-Distill-Llama-70B). (continued on the next page)

| GRPO Reasoning | To solve this problem, we need to follow the steps outlined in the question. Let's go through each step in detail. |
|---|---|
| | ### Step 1: Identify Articles with More Than 25 References |
| | First, we need to go through each provided article and determine which ones have more than 25 references. We'll count the words in the titles using spaces as delimiters to ensure accuracy. |
| | #### Article Titles and Reference Counts: |
| | 1. **Article Title:** "A SYSTEMATIC EVALUATION OF LARGE LANGUAGE MODELS OF CODE" |
| | - Words: Count spaces to determine words. |
| | "' A SYSTEMATIC EVALUATION OF LARGE LANGUAGE MODELS OF CODE "' |
| | Spaces: 8 words (A, SYSTEMATIC, EVALUATION, OF, LARGE, LANGUAGE, MODELS, OF, CODE) |
| | - Number of references: 27 (as given in the problem statement). |
| | This article has more than 25 references. |
| | 2. **Article Title:** "Reputation-based partner choice promotes cooperation in social networks" |
| | - Words: Count spaces to determine words. |
| | "' Reputation-based partner choice promotes cooperation in social networks "' |
| | Spaces: 9 words (Reputation-based, partner, choice, promotes, cooperation, in, social, networks). |
| | - Number of references: 26 (as given in the problem statement). |
| | This article has more than 25 references. |
| | `<...we cut it down to save space...>` |
| | ### Step 2: Determine the Smallest Number of Authors in Articles with More Than 25 References |
| | Now that we've identified the articles with more than 25 references, let's count the authors in each of these articles. |
| | 1. **Article Title:** "A SYSTEMATIC EVALUATION OF LARGE LANGUAGE MODELS OF CODE" |
| | - Authors: Frank F. Xu, Uri Alon, Graham Neubig, Vincent J. Hellendoorn (4 authors). |
| | 2. **Article Title:** "Reputation-based partner choice promotes cooperation in social networks" |
| | - Authors: Feng Fu, Christoph Hauert, Martin A. Nowak, Long Wang (4 authors). |
| | 3. **Article Title:** "Efficient Multimodal Sampling via Tempered Distribution Flow" |
| | - Authors: Yixuan Qiu, Xiao Wang (2 authors). |
| | 4. **Article Title:** "Ultralow contact resistance between semimetal and monolayer semiconductors" |
| | - Authors: Pin-Chun Shen, Cong Su, Yuxuan Lin, Ang-Sheng Chou, Chao-Ching Cheng, Ji-Hoon Park, Ming-Hui Chiu, Ang-Yu Lu, Hao-Ling Tang, Mohammad Mahdi Tavakoli, Gregory Pitner, Xiang Ji, Zhengyang Cai, Nannan Mao, Jiangtao Wang, Vincent Tung, Ju Li, Jeffrey Bokor, Alex Zettl, Chih-I Wu, Tomás Palacios, Lain-Jong Li, Jing Kong (11 authors). |
| | 5. **Article Title:** "Normal forms for singularities of pedal curves produced by non-singular dual curve germs in Sn" |
| | - Authors: Takashi Nishimura (1 author). |
| | ### Step 3: Identify the Smallest Number of Authors |
| | Among the articles that have more than 25 references, the smallest number of authors is \boxed{1}. |
| | ### Final Answer: |
| | The smallest number of authors for articles that have more than 25 references is \boxed{1}. |

Table 18: Example reasoning traces for different SQL query templates and their corresponding questions from different models (GRPO-based Qwen2.5-7B-Instruct-1M, Qwen3-4B-Thinking-2507 and DeepSeek-R1-Distill-Llama-70B). (continued on the next page)

| Qwen3 Reasoning | Let me go through all the articles to find those with more than 25 references and determine the smallest number of authors for those articles. |
|---|---|
| | First, I need to identify the number of references in each article. The references are listed at the end of each article, and I need to count them. |
| | Let's go through each article: |
| | 1. The first article is about "A SYSTEMATIC EVALUATION OF LARGE LANGUAGE MODELS OF CODE" by Frank F. Xu, Uri Alon, Graham Neubig, Vincent J. Hellendoorn. Let me count the references: - The reference count appears to be 25 (I can see the reference numbers go up to 25) - This article has 25 references, which is not more than 25. |
| | `<...we cut it down to save space...>` |
| | 7. The seventh article is "Normal forms for singularities of pedal curves produced by non-singular dual curve germs in Sn" by Takashi Nishimura. - The reference count appears to be 12 (I can see the references go up to 12) - This article has 12 references, which is not more than 25. |
| | So, from my analysis, the articles with more than 25 references are: 1. "Reputation-based partner choice promotes cooperation in social networks" - 57 references, 4 authors |
| | `<...we cut it down to save space...>` |
| | The smallest number of authors for articles with more than 25 references is 2 (from the "Efficient Multimodal Sampling..." article). |
| | Let me double-check the reference counts to be sure: |
| | 1. "Reputation-based partner choice promotes cooperation in social networks" - I counted 57 references (the references are numbered from 1 to 57) 2. "Task-Adaptive Angle Selection for Computed Tomography-Based Defect Detection" - I counted 27 references (the references are numbered from 1 to 27) 3. "Emergence of cooperation and evolutionary stability in finite populations" - I counted 38 references (the references are numbered from 1 to 38) 4. "Efficient Multimodal Sampling via Tempered Distribution Flow" - I counted 53 references (the references are numbered from 1 to 53) 5. "Ultralow contact resistance between semimetal and monolayer semiconductors" - I counted 56 references (the references are numbered from 1 to 56) |
| | All of these articles have more than 25 references, and the smallest number of authors is 2 (from the "Efficient Multimodal Sampling..." article). |
| | So the answer is 2. `</think>` |
| | To find the smallest number of authors for articles with more than 25 references, I need to: |
| | 1. Count the number of references in each article 2. Identify articles with more than 25 references 3. Find the smallest number of authors among those articles |
| | Let me analyze each article: |
| | 1. "A SYSTEMATIC EVALUATION OF LARGE LANGUAGE MODELS OF CODE" - 25 references (not more than 25) 2. "Reputation-based partner choice promotes cooperation in social networks" - 57 references (more than 25), 4 authors |
| | `<...we cut it down to save space...>` 7. "Normal forms for singularities of pedal curves produced by non-singular dual curve germs in Sn" - 12 references (not more than 25) |
| | The articles with more than 25 references are: - "Reputation-based partner choice promotes cooperation in social networks" (57 references, 4 authors) - "Task-Adaptive Angle Selection for Computed Tomography-Based Defect Detection" (27 references, 7 authors) - "Emergence of cooperation and evolutionary stability in finite populations" (38 references, 4 authors) - "Efficient Multimodal Sampling via Tempered Distribution Flow" (53 references, 2 authors) - "Ultralow contact resistance between semimetal and monolayer semiconductors" (56 references, 21 authors) |
| | The smallest number of authors for articles with more than 25 references is 2 (from "Efficient Multimodal Sampling via Tempered Distribution Flow"). |
| | `\boxed{2}` |

Table 18: Example reasoning traces for different SQL query templates and their corresponding questions from different models (GRPO-based Qwen2.5-7B-Instruct-1M, Qwen3-4B-Thinking-2507 and DeepSeek-R1-Distill-Llama-70B). (continued on the next page)

| DeepSeek-R1 Reasoning | Alright, I need to figure out how to answer the question about the smallest number of authors in an article with more than 25 references. Let me try to process this step by step.

First, I need to scan through the provided articles and look for the number of authors per article. The articles are in the text. So, I'll go through each paper and count the number of authors.

Looking at the first article, which is about a systematic evaluation of large language models for code, the authors are Frank F. Xu, Uri Alon, Roy Sadaka, Omer Levy, and Vincent J. Hellendoorn. So, that's 5 authors.

The second article is about the emergence of cooperation and evolutionary stability in finite populations. The authors are Martin A. Nowak, Akira Sasaki, Christine Taylor, and Drew Fudenberg. That's 4 authors.

The third article is about task-adaptive angle selection for CT-based defect detection. The authors are Tianyuan Wang, Virginia Florian, etc. So, that's 5 authors.

The fourth article is about efficient multimodal sampling via tempered distributions. The authors are Yixuan Qiu, Xiao Wang, and others. That's 2 authors.

The fifth article is about ultralow contact resistance between semimetal and monolayer semiconductors. The authors are Pin-Chun Shen, Cong Su, Yuxuan Lin, Ang-Sheng Chou, Chao-Ching Cheng, etc. Wait, how many authors are in this paper? Let me count them. There are 12 authors. So, that's 12 authors.

The sixth article is about normal forms for singularities of pedal curves. The author is Takashi Nishimura. Single author.

The seventh article is about the geometry of singularities. The author is Takashi Nishimura.

The eighth article is about the blowup type maps. The author is Takashi Nishimura.

So, reviewing all, I need to find the article with more than 25 references. Now, the question is: what is the smallest number of authors in an article with more than 25 references.

Looking back, the first article has 5 authors, the second has 4 authors, the third has 5 authors, the fourth has 2 authors, the fifth has 12 authors, the sixth has 1 author, the seventh has 1 author, and the eighth has 1 author.

Wait, but the fifth article has 12 authors, which is more than 25 references. So, if an article has 12 authors, that's the number. But wait, the question is about the smallest number of authors for articles with more than 25 references.

Wait, no, the question is about the smallest number of authors in an article with more than 25 references. So, the fifth article has 12 authors.

But wait, what's the smallest number of authors for an article with more than 25 references? The fifth article has 12 authors, which is more than 25. So, the smallest number is 1 author, but that's for 25 references. But the question is about the number of authors, not the number of references.

Wait, no, the question is about the number of authors in the articles with more than 25 references. So, I need to find the article with the smallest number of authors and more than 25 references.

Looking back, the fifth article has 12 authors, but other articles have more. So, perhaps the fifth article has the largest number of authors. But the question is to find the smallest number of authors in an article with more than 25 references.

Wait, the question is to find the smallest number of authors in an article with more than 25 references.

Wait, no, the question is to find the smallest number of authors in an article with more than 25 references. So, if an article has more than 25 references, we need to find the smallest number of authors. For example, the fifth article has 12 authors, which is more than 25 authors.

But the question is about the smallest number of authors in an article with more than 25 references. So, the answer is 12 authors.

Wait, no, the question is about the number of authors in the articles, not the number of references. So, the fifth article has 12 authors, which is more than 25 authors.

But the question is to find the smallest number of authors |

| Template 1 | `SELECT MIN(author_count) FROM articles WHERE reference_count > {reference-count}` |

Table 18: Example reasoning traces for different SQL query templates and their corresponding questions from different models (GRPO-based Qwen2.5-7B-Instruct-1M, Qwen3-4B-Thinking-2507 and DeepSeek-R1-Distill-Llama-70B). 

| Question Context 512K | What is the smallest number of authors for articles that cite more than 84 references? |
|---|---|
| Gold Answer | 2 |
| GRPO Reasoning | To solve this problem, we need to follow these steps: 
 1. Identify articles that cite more than 84 references. 
 2. Determine the number of authors in each qualifying article. 
 3. Identify the smallest number of authors among these articles. 
 4. Provide the final answer in the required format. 
 Let's go through each step in detail. 
 ### Step 1: Identify articles that cite more than 84 references 
 First, we will examine each provided article to determine which ones have more than 84 references. 
 1. **Article Title:** "Macroscopic model and statistical model to characterize electromagnetic information of a digital coding metasurface" 
 Number of words in title: Counting spaces, we have "Macroscopic model and statistical model to characterize electromagnetic information of a digital coding metasurface" → Words are: Macroscopic, model, and, statistical, model, to, characterize, electromagnetic, information, of, a, digital, coding, metasurface. (12 words). 
 Number of references: 85 (as stated in the problem). 
 2. **Article Title:** "Worst-case VaR and robust portfolio optimization with interval random uncertainty set" 
 Number of words in title: "Worst-case VaR and robust portfolio optimization with interval random uncertainty set" → Words are: Worst-case, VaR, and, robust, portfolio, optimization, with, interval, random, uncertainty, set. (11 words). 
 Number of references: 84 (as stated in the problem). 
 `<...we cut it down to save space...>` 
 Now we have identified the following articles that cite more than 84 references: 
 - "Macroscopic model and statistical model to characterize electromagnetic information of a digital coding metasurface" (85 references). 
 - "Worst-case VaR and robust portfolio optimization with interval random uncertainty set" (84 references). 
 - "Direct Transmission of Digital Message via Programmable Coding Metasurface" (82 references). 
 - "Bayesian experimental design for linear elasticity" (84 references). 
 - "Metaverse for Social Good: A University Campus Prototype" (84 references). 
 - "Alloying effect-induced electron polarization drives nitrate electroreduction to ammonia" (84 references). 
 ### Step 2: Determine the number of authors in each qualifying article 
 Next, we will count the authors in each of these identified articles. 
 1. **"Macroscopic model and statistical model to characterize electromagnetic information of a digital coding metasurface"** 
 Authors listed: Rui Wen Shao, Wensheng Gan (listed as corresponding author), Philip S. Yu (listed as corresponding author). 
 Number of authors: 3. 
 `<...we cut it down to save space...>` 
 ### Step 3: Identify the smallest number of authors among these articles 
 Among the identified articles that cite more than 84 references, the smallest number of authors is 3. 
 ### Final Answer 
 The smallest number of authors for articles that cite more than 84 references is `\boxed{3}`. |

| Template 1 | `SELECT MIN(author_count) FROM articles WHERE reference_count > {reference-count}` |
|---|---|
| Question Context 1M | What is the smallest number of authors for articles that cite more than 51 references? |
| Gold Answer | 2 |

Table 18: Example reasoning traces for different SQL query templates and their corresponding questions from different models (GRPO-based Qwen2.5-7B-Instruct-1M, Qwen3-4B-Thinking-2507 and DeepSeek-R1-Distill-Llama-70B). (continued on the next page)

| GRPO Reasoning | To determine the smallest number of authors for articles that cite more than 51 references, we need to follow these steps:
1. Identify the smallest number of authors for each article. |
|---|---|

| Template 2 | `SELECT COUNT(DISTINCT article_id) FROM article_author WHERE author_name = {author-name} AND author_position = 0` |
|---|---|
| Question Context 64K | How many different articles has Shi Jin been listed as the first author of? |
| Gold Answer | 0 |
| GRPO Reasoning | To determine how many different articles Shi Jin has been listed as the first author of, we need to go through each article provided in the documents and check his position as an author. Let's examine each article one by:
1. **Broadband diffusion of terahertz waves by multi-bit coding metasurfaces**:
- Authors: Li-Hua Gao, Qiang Cheng, Jing Yang, Shao-Jie Ma, Jie Zhao, Shuo Liu, Hai-Bing Chen, Qiong He, Wei-Xiang Jiang, Hui-Feng Ma, Qi-Ye Wen, Lan-Ju Liang, Biao-Bing Jin, Wei-Wei Liu, Lei Zhou, Jian-Quan Yao, Pei-Heng Wu, Tie-Jun Cui
- Shi Jin is not listed as the first author here.
2. **Large Intelligent Surface-Assisted Wireless Communication Exploiting Statistical CSI**:
- Authors: Yu Han, Wankai Tang, Shi Jin, Chao-Kai Wen, Xiaoli Ma
- Shi Jin is listed as the fourth author but not the first.
3. **Intelligent Reflecting Surface Enhanced Wireless Network: Joint Active and Passive Beamforming Design**:
- Authors: Qingqing Wu, Rui Zhang
- Shi Jin is not listed as an author here.
4. **Wireless Communication Based on Information Metasurfaces**:
- Authors: Jun Yan Dai, Wankai Tang, Ming Zheng Chen, Chi Hou Chan, Qiang Cheng, Shi Jin, Tie Jun Cui
- Shi Jin is listed as the seventh author but not the first.
After examining all provided articles, Shi Jin has not been listed as the first author in any of them.
Thus, the number of different articles Shi Jin has been listed as the first author of is 0.
\boxed{0} |

Table 18: Example reasoning traces for different SQL query templates and their corresponding questions from different models (GRPO-based Qwen2.5-7B-Instruct-1M, Qwen3-4B-Thinking-2507 and DeepSeek-R1-Distill-Llama-70B). (continued on the next page)

| Qwen3 Reasoning | I need to carefully go through the provided scientific articles to find how many different articles Shi Jin has been listed as the first author of. |
|---|---|
| | First, I'll scan through all the articles to find Shi Jin as an author. |
| | Let me go through each article: |
| | 1. First article: "Articles: [article title] Li-Hua Gao, Qiang Cheng, Jing Yang, Shao-Jie Ma, Jie Zhao, Shuo Liu, Hai-Bing Chen, Qiong He, Wei-Xiang Jiang, Hui-Feng Ma, Qi-Ye Wen, Lan-Ju Liang, Biao-Bing Jin, Wei-Wei Liu, Lei Zhou, Jian-Quan Yao, Pei-Heng Wu and Tie-Jun Cui" |
| | This article doesn't list Shi Jin as an author. |
| | `<...we cut it down to save space...>` |
| | Let me check the references section for any papers where Shi Jin is listed as the first author. |
| | Looking at the references section, I see: |
| | [32] X. Yang, C.-K. Wen, and S. Jin, "MIMO detection for reconfigurable intelligent surface-assisted millimeter wave systems," IEEE J. Sel. Areas Commun., vol. 38, no. 8, pp. 1777–1792, Aug. 2020. |
| | This is a reference to a paper with Shi Jin as a co-author, but not as the first author. |
| | [33] C. Huang, A. Zappone, G. C. Alexandropoulos, M. Debbah, and C. Yuen, "Reconfigurable intelligent surfaces for energy efficiency in wireless communication," IEEE Trans. Wireless Commun., vol. 18, no. 8, pp. 4157–4170, Aug. 2019. |
| | This doesn't list Shi Jin as an author. |
| | `<...we cut it down to save space...>` |
| | Let me check the other references more carefully. |
| | I see in reference [88]: "Y. Han, W. Tang, S. Jin, C.-K. Wen, and X. Ma, "Large intelligent surface-assisted wireless communication exploiting statistical CSI," IEEE Trans. Veh. Technol., vol. 68, no. 8, pp. 8238–8242, Aug. 2019." |
| | This is a paper where Shi Jin is listed as an author (S. Jin), but not as the first author. The first author is Y. Han. |
| | `<...we cut it down to save space...>` |
| | Let me double-check the references section for any papers where Shi Jin might be listed as the first author. |
| | I don't see any paper in the provided text where Shi Jin is listed as the first author. Shi Jin is listed as a co-author in at least two papers (the second and fourth articles), but never as the first author. |
| | So the answer is 0. |
| | Wait, I need to be careful. Let me check the reference list again. |
| | `<...we cut it down to save space...>` |
| | Actually, let me check the reference list again for any paper where Shi Jin might be listed as the first author. |
| | In the references, I see: [88] Y. Han, W. Tang, S. Jin, C.-K. Wen, and X. Ma, "Large intelligent surface-assisted wireless communication exploiting statistical CSI," IEEE Trans. Veh. Technol., vol. 68, no. 8, pp. 8238–8242, Aug. 2019. |
| | This shows Shi Jin as the third author (S. Jin), not the first author. |
| | So I think the answer is 0. |
| | `<...we cut it down to save space...>` |
| | After a thorough examination of all the provided articles and their author lists, I found no instances where Shi Jin is listed as the first author of any article in the provided text. The answer is 0. |
| | `\boxed{0}` |

Table 18: Example reasoning traces for different SQL query templates and their corresponding questions from different models (GRPO-based Qwen2.5-7B-Instruct-1M, Qwen3-4B-Thinking-2507 and DeepSeek-R1-Distill-Llama-70B). (continued on the next page)

| DeepSeek-R1 Reasoning | Alright, I need to figure out how many different articles Shi Jin has been listed as the first author of, based on the provided articles. Let me go through each article one by one to check the author lists. |
|---|---|
| | Looking at the first article titled "Broadband diffusion of terahertz waves by multi-bit coding metasurfaces," the authors are Li-Hua Gao, Qiang Cheng, Jing Yang, and several others. Shi Jin isn't the first author here. |
| | Next, the second article is about "Large Intelligent Surface-Assisted Wireless Communication Exploiting Statistical CSI." The authors are Yu Han, Wankai Tang, Shi Jin, Chao-Kai Wen, and Xiaoli Ma. Shi Jin is the third author, not the first. |
| | The third article, "Wireless Communication Based on Information Metasurfaces," lists Jun Yan Dai, Wankai Tang, Ming Zheng Chen, Chi Hou Chan, Qiang Cheng, Shi Jin, and Tie Jun Cui as authors. Shi Jin is the sixth author here. |
| | The fourth article is "Intelligent Reflecting Surface Enhanced Wireless Network: Joint Active and Passive Beamforming Design" by Qingqing Wu and Rui Zhang. Shi Jin isn't an author here. |
| | So, after checking all the provided articles, I don't see any where Shi Jin is listed as the first author. Therefore, the number of articles where Shi Jin is the first author is zero. </think> |
| | The number of different articles Shi Jin has been listed as the first author of is 0. |
| | \boxed{0} |

| Template 2 | `SELECT COUNT(DISTINCT article_id) FROM article_author WHERE author_name = {author-name} AND author_position = 0` |
|---|---|
| Question Context 128K | How many distinct articles has Tang been the first author of? |
| Gold Answer | 1 |

Table 18: Example reasoning traces for different SQL query templates and their corresponding questions from different models (GRPO-based Qwen2.5-7B-Instruct-1M, Qwen3-4B-Thinking-2507 and DeepSeek-R1-Distill-Llama-70B). (continued on the next page)

| GRPO Reasoning | To solve this problem, we need to follow a systematic approach by analyzing each article provided in the documents. Let's go through each step in detail. |
|---|---|
| | ### Step 1: Identify Articles Written by Tang as the First Author |
| | First, we need to examine each article to determine if Tang is listed as the first author. We will go through each article one by: |
| | #### Article 1: "The humping phenomenon during high speed gas metal arc welding" |
| | Authors listed: T. C. Nguyen, D. C. Weckman, D. A. Johnson, H. W. Kerr (D. C. Weckman is the corresponding author). |
| | Tang is not listed as an author in this article. |
| | #### Article 2: "Physics of Humping Formation in Laser Powder Bed Fusion" |
| | Authors listed: C. Tang, K. Q. Le, C. H. Wong (C. Tang is the corresponding author). |
| | Tang is listed as the first author in this article. |
| | #### Article 3: "Keyhole fluctuation and pore formation mechanisms during laser powder bed fusion additive manufacturing" Authors listed: Yuze Huang [1,2], Tristan G. Fleming [3], Samuel J. Clark [1,2,4], Sebastian Marussi [1,2], Kamel Fezzaa [4], Jeyan Thiyagalingam [5], Chu Lun Alex Leung [1,2], Peter D. Lee [1,2] (Yuze Huang is the first author). |
| | Tang is not listed as an author in this article. |
| | #### Article 4: "Unveiling mechanisms and onset threshold of humping in high-speed laser welding" |
| | Authors listed: Zen-Hao Lai [1], Siguang Xu [2], Samuel J. Clark [3], Kamel Fezzaa [3], Jingjing Li [1,4] (Zen-Hao Lai is the first author). |
| | Tang is not listed as an author in this article. |
| | #### Article 5: "Understanding of humping formation and suppression mechanisms using the numerical simulation" |
| | Authors listed: Dongsheng Wu [a,b,c], Xueming Hua [a,b,*], Dingjian Ye [a,b], Fang Li [a,b] (Dongsheng Wu is the first author). |
| | Tang is not listed as an author in this article. |
| | #### Article 6: "Investigation of the humping formation in the high power and high speed laser welding" |
| | Authors listed: Yuewei Ai [a,b], Ping Jia [a,b,*], Chunming Wang [b], Gaoyang Mi [b], Shaoning Geng [a], Wei Liu [a], Chu Han [a] (Yuewei Ai is the first author). |
| | Tang is not listed as an author in this article. |
| | ### Step 2: Count Distinct Articles Written by Tang as the First Author |
| | From the analysis above, only one article lists Tang as the first author: |
| | - "Physics of Humping Formation in Laser Powder Bed Fusion" by C. Tang, K. Q. Le, C. H. Wong. |
| | Thus, Tang has been the first author in exactly one distinct article. |
| | ### Final Answer |
| | The number of distinct articles Tang has been the first author of is \boxed{1}. |

Table 18: Example reasoning traces for different SQL query templates and their corresponding questions from different models (GRPO-based Qwen2.5-7B-Instruct-1M, Qwen3-4B-Thinking-2507 and DeepSeek-R1-Distill-Llama-70B). (continued on the next page)

| Qwen3 Reasoning | Let me go through the provided articles to identify all the articles where Tang is the first author.
First, I'll look for the article with "Tang" as the first author.
Looking through the articles, I find this one:
Physics of Humping Formation in Laser Powder Bed Fusion
C. Tang , K. Q. Le, C. H. Wong
a Singapore Centre for 3D Printing, School of Mechanical and Aerospace Engineering, Nanyang Technological University, 50 Nanyang Avenue, Singapore 639798, Singapore
b Independent Researcher, Singapore
This is the first article where Tang is listed as the first author (with  indicating the corresponding author).
Let me look for any other articles where Tang is the first author.
I don't see any other articles where "Tang" is listed as the first author. The other articles have different first authors.
Let me double-check:
1. The article about "The humping phenomenon during high speed gas metal arc welding" has author T. C. Nguyen as the first author.
`<...we cut it down to save space...>`
7. The article about "Calculation of laser absorption by metal powders in additive manufacturing" has C. D. Boley as the first author.
So, based on the provided articles, Tang is only the first author of one article: "Physics of Humping Formation in Laser Powder Bed Fusion".
Let me check if there's any other article where Tang might be listed as the first author.
I don't see any other article where "Tang" appears as the first author.
The article "Physics of Humping Formation in Laser Powder Bed Fusion" is the only one where Tang is listed as the first author.
Therefore, the answer is 1.
Let me make sure I didn't miss any other articles where Tang might be the first author.
I've gone through all the articles and can't find any other article where Tang is the first author.
So, Tang has been the first author of 1 distinct article.
Actually, I need to be careful about the formatting. The question asks for "how many distinct articles has Tang been the first author of".
Let me double-check if there's another article where Tang is listed as the first author.
In the "Physics of Humping Formation in Laser Powder Bed Fusion" article, Tang is listed as the first author with a (corresponding author) designation.
I don't see any other article where Tang is listed as the first author.
So, the answer is 1. `</think>`
I need to find how many distinct articles Tang has been the first author of.
Let me go through each article to check for Tang as the first author:
1. "The humping phenomenon during high speed gas metal arc welding" - First author is T. C. Nguyen
`<...we cut it down to save space...>`
6. "Investigation of the humping formation in the high power and high speed laser welding" - First author is Yuewei Aia
7. "Physics of Humping Formation in Laser Powder Bed Fusion" - First author is C. Tang (with  indicating corresponding author)
8. "Calculation of laser absorption by metal powders in additive manufacturing" - First author is C. D. Boley
I found only one article where Tang is listed as the first author: "Physics of Humping Formation in Laser Powder Bed Fusion".
So Tang has been the first author of 1 distinct article.
`\boxed{1}` |
|---|---|

Table 18: Example reasoning traces for different SQL query templates and their corresponding questions from different models (GRPO-based Qwen2.5-7B-Instruct-1M, Qwen3-4B-Thinking-2507 and DeepSeek-R1-Distill-Llama-70B). (continued on the next page)

| DeepSeek-R1 Reasoning | Okay, I need to determine how many distinct articles Tang has been the first author of. I'll start by looking through each article to see if Tang is the first author. |
|---|---|
| | First, I'll go through each article one by one. |
| | 1. The first article is about the humping phenomenon in high-speed gas metal arc welding. The authors are T.C. Nguyen, D.C. Weckman, D.A. Johnson, and H.W. Kerr. Tang is not an author here. |
| | 2. The second article discusses laser powder bed fusion and humping. The authors are C. Tang, K.Q. Le, C.H. Wong, and others. Tang is the first author here, so that's one article. |
| | 3. The third article is about humping in laser welding. The authors are Yuze Huang, Tristan G. Fleming, Samuel J. Clark, Kamel Fezzaa, and others. Tang is not an author here. |
| | 4. The fourth article is about humping formation and suppression in high-speed laser welding. The authors are Zen-Hao Lai, Siguang Xu, Samuel J. Clark, and others. Tang is not an author here. |
| | 5. The fifth article is about the humping phenomenon in high-speed GMAW. The authors are Dongsheng Wu, Xueming Hua, Dingjian Ye, and others. Tang is not an author here. |
| | 6. The sixth article is about laser absorption by metal powders in additive manufacturing. The authors are C.D. Boley, S.A. Khairallah, and A.M. Rubenchik. Tang is not an author here. |
| | So, from the list, Tang is the first author in only the second article. |
| | Wait, but I should check all the articles again to make sure I didn't miss any. |
| | Looking again: |
| | - The second article has Tang as the first author. - The other articles have different first authors like Yuze Huang, Zen-Hao Lai, Dongsheng Wu, and others, but none of these are Tang. |
| | So, Tang has been the first author in only one article. `</think>` |
| | The number of distinct articles that Tang has been the first author of is 1. |
| | `\boxed{1}` |

| Template 2 | `SELECT COUNT(DISTINCT article_id) FROM article_author WHERE author_name = {author-name} AND author_position = 0` |
|---|---|
| Question Context 512K | How many different articles was Connor M. Brennan the first author of? |
| Gold Answer | 1 |

Table 18: Example reasoning traces for different SQL query templates and their corresponding questions from different models (GRPO-based Qwen2.5-7B-Instruct-1M, Qwen3-4B-Thinking-2507 and DeepSeek-R1-Distill-Llama-70B). (continued on the next page)

| GRPO Reasoning | To solve this problem, we need to follow the steps outlined in the question. Let's go through each step carefully and determine the final answer.
### Step 1: Identify Relevant Articles First, we need to scan through the provided articles to identify which ones have Connor M. Brennan as the first author. This will help us determine how many different articles he authored.
#### Article Titles and Authors: 1. **"The asymmetric effects of monetary policy on stock price bubbles"** by Christophe Blot, Paul Hubert, Fabien Labondance.
2. **"Deconstructing monetary policy surprises: the role of information shocks"** by Marek Jarociński, Peter Karadi.
3. **"Nominal Rigidities and the Dynamic Effects of a Shock to Monetary Policy"** by Lawrence J. Christiano, Martin Eichenbaum, Charles Evans.
4. **"Risk Appetite and the Risk-Taking Channel of Monetary Policy"** by Michael D. Bauer, Ben S. Bernanke, Eric Milstein.
5. **"The Macroeconomic Effects of the Federal Reserve's Conventional and Unconventional Monetary Policies"** by Eric T. Swanson.
6. **"Federal Reserve Information and the Behavior of Interest Rates"** by Christina D. Romer, David H. Romer.
7. **"A Reassessment of Monetary Policy Surprises and High-Frequency Identification"** by Michael D. Bauer, Eric T. Swanson.
8. **"An Alternative Explanation for the 'Fed Information Effect'"** by Michael D. Bauer, Eric T. Swanson.
9. **"Monetary Policy Shocks: Data or Methods?"** by Connor M. Brennan, Margaret M. Jacobson, Christian Matthes, Todd B. Walker.
10. **"High Asymmetric Effects of Monetary Policy on Stock Price Bubbles"** by Christophe Blot, Paul Hubert, Fabien Labondance (Note: This is a duplicate title and should be ignored).
From this list, only one article has Connor M. Brennan as the first author:
- **"Monetary Policy Shocks: Data or Methods?"** by Connor M. Brennan, Margaret M. Jacobson, Christian Matthes, Todd B. Walker.
### Step 2: Verify Connor M. Brennan as the First Author Now, let's confirm that Connor M. Brennan is indeed the first author of the identified article:
- The article title is "Monetary Policy Shocks: Data or Methods?"
- The authors listed are Connor M. Brennan, Margaret M. Jacobson, Christian Matthes, Todd B. Walker.
- Connor M. Brennan is listed first, confirming he is the first author.
### Step 3: Count Words in the Title Using Spaces Next, we need to count the words in the title "Monetary Policy Shocks: Data or Methods?" using spaces as delimiters. Let's list each word:
1. Monetary
2. Policy
3. Shocks:
4. Data
5. or
6. Methods?
There are 6 words in the title.
### Final Answer
Connor M. Brennan was the first author of \boxed{1} different article. |
|---|---|

| Template 2 | `SELECT COUNT(DISTINCT article_id) FROM article_author WHERE author_name = {author-name} AND author_position = 0` |
|---|---|
| Question Context 1M | How many different articles has Saurabh Bodhe authored as the first author? |
| Gold Answer | 0 |
| GRPO Reasoning | To answer the question, let's break down the process step by step. |

Table 18: Example reasoning traces for different SQL query templates and their corresponding questions from different models (GRPO-based Qwen2.5-7B-Instruct-1M, Qwen3-4B-Thinking-2507 and DeepSeek-R1-Distill-Llama-70B). (continued on the next page)

| Template 3 | SELECT COUNT(DISTINCT article_id) FROM article_author WHERE author_name = {author-name} |
|---|---|
| Question Context 64K | How many different articles has Feng Miao authored? |
| Gold Answer | 1 |
| GRPO Reasoning | To determine how many different articles Feng Miao has authored, we need to go through each article provided and identify Feng Miao's involvement in each one. Let's analyze each article one by step. 

 ### Article 1: "Hopping transport through defect-induced localized states in molybdenum disulphide" 
 Authors listed: Hao Qiu, Tao Xu, Zilu Wang, Wei Ren, Haiyan Nan, Zhenhua Ni, Qian Chen, Shijun Yuan, Feng Miao, Fengqi Song, Gen Long, Yi Shi, Litao Sun, Jinlan Wang, Xinran Wang. 
 Feng Miao is listed as an author in this article. 
 ### Article 2: "Approaching the quantum limit in two-dimensional semiconductor contacts" 
 Authors listed: Weisheng Li, Xiaoshu Gong, Zhihao Yu, Liang Ma, Wenjie Sun, Si Gao, Çağıl Köroğlu, Wenfeng Wang, Lei Liu, Taotao Li, Hongkai Ning, Dongxu Fan, Yifei Xu, Xuecou Tu, Tao Xu, Litao Sun, Wenhui Wang, Junpeng Lu, Zhenhua Ni, Jia Li, Xidong Duan, Peng Wang, Yuefeng Nie, Hao Qiu, Yi Shi, Eric Pop, Jinlan Wang, Xinran Wang. 
 Feng Miao is not listed as an author in this article. 
 ### Article 3: "Origins of genuine Ohmic van der Waals contact between indium and MoS2" 
 Authors listed: Bum-Kyu Kim, Tae-Hyung Kim, Dong-Hwan Choi, Hanul Kim, Kenji Watanabe, Takashi Taniguchi, Heesuk Rho, Ju-Jin Kim, Yong-Hoon Kim, Myung-Ho Bae. 
 Feng Miao is not listed as an author in this article. 
 ### Article 4: "Van der Waals contacts between three-dimensional metals and two-dimensional semiconductors" 
 Authors listed: Yan Wang, Jong Chan Kim, Ryan J. Wu, Jenny Martinez, Xiuju Song, Jieun Yang, Fang Zhao, Andre Mkhoyan, Hu Young Jeong, Manish Chhowalla. 
 Feng Miao is not listed as an author in this article. 
 ### Conclusion 
 Based on the analysis, Feng Miao has authored only one article, which is "Hopping transport through defect-induced localized states in molybdenum disulphide." 
 Thus, the number of different articles Feng Miao has authored is \boxed{1}. |

Table 18: Example reasoning traces for different SQL query templates and their corresponding questions from different models (GRPO-based Qwen2.5-7B-Instruct-1M, Qwen3-4B-Thinking-2507 and DeepSeek-R1-Distill-Llama-70B). (continued on the next page)

| | |
|---|---|
| Qwen3 Reasoning | I need to find out how many different articles Feng Miao has authored based on the provided scientific articles. |
| | Let me go through each article and check if Feng Miao is listed as an author. |
| | First article (Hopping transport through defect-induced localized states in molybdenum disulphide): |
| | This is a Nature Communications article from 2013. The authors are listed as: Hao Qiu, Tao Xu, Zilu Wang, Wei Ren, Haiyan Nan, Zhenhua Ni, Qian Chen, Shijun Yuan, Feng Miao, Fengqi Song, Gen Long, Yi Shi, Litao Sun, Jinlan Wang & Xinran Wang |
| | Feng Miao is listed as author #5 in this article. |
| | `<...we cut it down to save space...>` |
| | Fourth article (Van der Waals contacts between three-dimensional metals and two-dimensional semiconductors): |
| | This is a Nature article from 2019. The authors are listed as: Yan Wang[1,2], Jong Chan Kim[3] , Ryan J. Wu[4], Jenny Martinez[5], Xiuju Song[2,6], Jieun Yang[1,2], Fang Zhao[7], Andre Mkhoyan[4], Hu Young Jeong[3] & Manish Chhowalla[1,2,6]* |
| | I don't see Feng Miao listed as an author in this article. |
| | So from the provided articles, I can see that Feng Miao is listed as an author in only one article (the first one: "Hopping transport through defect-induced localized states in molybdenum disulphide"). |
| | Let me double-check if there are any other articles in the provided text that I might have missed. |
| | I've checked all four articles listed in the provided text. In the first article, Feng Miao is listed as an author (with a superscript 5 indicating his affiliation). In the other three articles, Feng Miao is not listed as an author. |
| | Therefore, based on the provided information, Feng Miao has authored 1 different article. |
| | Actually, let me check the first article more carefully. The article title is "Hopping transport through defect-induced localized states in molybdenum disulphide" and it's a Nature Communications article from 2013. Feng Miao is listed as the 9th author (Feng Miao). |
| | I don't see any other articles where Feng Miao is listed as an author in the provided text. So the answer is 1. ¡/think¿ |
| | To determine how many different articles Feng Miao has authored based on the provided text, I need to carefully examine each article's author list. |
| | First article: "Hopping transport through defect-induced localized states in molybdenum disulphide" (Nature Communications, 2013) - Authors: Hao Qiu, Tao Xu, Zilu Wang, Wei Ren, Haiyan Nan, Zhenhua Ni, Qian Chen, Shijun Yuan, **Feng Miao**, Fengqi Song, Gen Long, Yi Shi, Litao Sun, Jinlan Wang & Xinran Wang - Feng Miao is listed as author #9 with affiliation 5 |
| | `<...we cut it down to save space...>` |
| | Fourth article: "Van der Waals contacts between three-dimensional metals and two-dimensional semiconductors" (Nature, 2019) - Authors: Yan Wang, Jong Chan Kim, Ryan J. Wu, Jenny Martinez, Xiuju Song, Jieun Yang, Fang Zhao, Andre Mkhoyan, Hu Young Jeong & Manish Chhowalla - Feng Miao is not listed as an author |
| | After examining all four articles in the provided text, I can see that Feng Miao is only listed as an author in the first article ("Hopping transport through defect-induced localized states in molybdenum disulphide"). He is not listed as an author in any of the other articles. |
| | Therefore, based on the provided information, Feng Miao has authored 1 different article. |
| | `\boxed{1}` |

Table 18: Example reasoning traces for different SQL query templates and their corresponding questions from different models (GRPO-based Qwen2.5-7B-Instruct-1M, Qwen3-4B-Thinking-2507 and DeepSeek-R1-Distill-Llama-70B). (continued on the next page)

| DeepSeek-R1 Reasoning | Alright, I need to figure out how many different articles Feng Miao has authored based on the provided information. Let's start by looking through each article to find instances where Feng Miao is listed as an author. |
|---|---|
| | First, I'll go through each article one by one. |
| | 1. The first article titled "Hopping transport through defect-induced localized states in molybdenum disulphide" has several authors. Looking at the author list, I see "Feng Miao" and "Feng Miao[1]". Wait, that might be the same person listed twice with different affiliations. So, in this article, Feng Miao is an author. |
| | 2. The second article, "Approaching the quantum limit in two-dimensional semiconductor contacts," has a long list of authors. Scanning through, I see "Feng Miao, Feng Miao" again, which seems to be the same person with different affiliations. So, Feng Miao is an author here as well. |
| | 3. The third article, "Origins of genuine Ohmic van der Waals contact between indium and MoS2," doesn't list Feng Miao as an author. The authors mentioned are Bum-Kyu Kim, Tae-Hyung Kim, Dong-Hwan Choi, Hanul Kim, Kenji Watanabe, Takashi Taniguchi, Heesuk Rho, Ju-Jin Kim, Yong-Hoon Kim, and Myung-Ho Bae. So, Feng Miao isn't in this one. |
| | 4. The fourth article, "Van der Waals contacts between three-dimensional metals and two-dimensional semiconductors," also doesn't include Feng Miao in the author list. The authors are Yan Wang, Jong Chan Kim, Ryan J. Wu, Jenny Martinez, Xiuju Song, Jieun Yang, Fang Zhao, Andre Mkhoyan, Hu Young Jeong, and Manish Chhowalla. No mention of Feng Miao. |
| | So, from the four articles provided, Feng Miao is an author in the first two. However, in each of those, he's listed twice with different affiliations, but that's the same person. Therefore, Feng Miao has authored two distinct articles in the provided list. |
| | Wait, but looking again, the first article has Feng Miao listed as Feng Miao and Feng Miao. That's the same person with different affiliations, so it's one article. The second article has Feng Miao listed as Feng Miao and Feng Miao, again same person, so that's another single article. Therefore, Feng Miao has authored two different articles in total. </think> |
| | Feng Miao has authored 2 different articles. |
| | Answer: 2 |

| Template 3 | `SELECT COUNT(DISTINCT article_id) FROM article_author WHERE author_name = {author-name}` |
|---|---|
| Question Context 128K | How many different articles has JIAYI SUN authored? |
| Gold Answer | 2 |

Table 18: Example reasoning traces for different SQL query templates and their corresponding questions from different models (GRPO-based Qwen2.5-7B-Instruct-1M, Qwen3-4B-Thinking-2507 and DeepSeek-R1-Distill-Llama-70B). (continued on the next page)

| GRPO Reasoning | To solve this problem, we need to follow these steps: |
|---|---|
| | 1. Identify articles authored by JIAYI SUN. |
| | 2. Count the number of distinct articles authored by JIAYI SUN. |
| | 3. Provide the final answer in the required format. |
| | Let's go through each step in detail: |
| | ### Step 1: Identify articles authored by JIAYI SUN |
| | From the provided articles, we need to examine each one to determine if JIAYI SUN is listed as an author. Let's review each article: |
| | 1. **Article Title:** "Solipsis: A Decentralized Architecture for Virtual Environments" |
| | Authors listed: Davide Frey, Jérôme Royan, Romain Piegay, Anne-Marie Kermarrec, Emmanuelle Anceaume, Fabrice Le Fessant, and Jiayi Sun. |
| | JIAYI SUN is present here. |
| | 2. **Article Title:** "Metaverse: Survey, Applications, Security, and Opportunities" |
| | Authors listed: Jiayi Sun, Wensheng Gan, Han-Chieh Chao, and Philip S. Yu. |
| | JIAYI SUN is present here. |
| | 3. **Article Title:** "Big Data Meets Metaverse: A Survey" |
| | Authors listed: Jiayi Sun, Wensheng Gan, Zefeng Chen, Junhui Li, and Philip S. Yu. |
| | JIAYI SUN is present here. |
| | 4. **Article Title:** "A Dynamic Resource Allocation Framework for Synchronizing Metaverse with IoT Service and Data" |
| | Authors listed: Yue Han, Dusit Niyato, Cyril Leung, Chunyan Miao, and Jiayi Sun. |
| | JIAYI SUN is present here. |
| | Thus, JIAYI SUN has authored four distinct articles. |
| | ### Step 2: Count the number of distinct articles authored by JIAYI SUN |
| | From the analysis above, we have identified the following articles authored by JIAYI SUN: |
| | 1. "Solipsis: A Decentralized Architecture for Virtual Environments" |
| | 2. "Metaverse: Survey, Applications, Security, and Opportunities" |
| | 3. "Big Data Meets Metaverse: A Survey" |
| | 4. "A Dynamic Resource Allocation Framework for Synchronizing Metaverse with IoT Service and Data" |
| | So, there are \boxed{4} distinct articles authored by JIAYI SUN. |
| | ### Final Answer |
| | The number of different articles authored by JIAYI SUN is \boxed{4}. |

Table 18: Example reasoning traces for different SQL query templates and their corresponding questions from different models (GRPO-based Qwen2.5-7B-Instruct-1M, Qwen3-4B-Thinking-2507 and DeepSeek-R1-Distill-Llama-70B). (continued on the next page)

| Qwen3 Reasoning | I need to carefully search through the provided articles to find all the articles authored by JIAYI SUN. |
|---|---|
| | Let me go through each article one by one: |
| | 1. First article: "Solipsis: A Decentralized Architecture for Virtual Environments" - Author is Davide Frey, Jérôme Royan, Romain Piegay, Anne-Marie Kermarrec, Emmanuelle Anceaume, Fabrice Le Fessant. Not JIAYI SUN. |
| | `<...we cut it down to save space...>` |
| | Let me check if there are any other articles by JIAYI SUN: |
| | In the "Big Data Meets Metaverse: A Survey" article, the authors are listed as "Jiayi Sun, Wensheng Gan, Zefeng Chen, Junhui Li, Philip S. Yu, Fellow, IEEE". The name is written as "Jiayi Sun" (with a space between "Jiayi" and "Sun"). |
| | In the "Metaverse: Survey, Applications, Security, and Opportunities" article, the authors are listed as "JIAYI SUN, Wensheng Gan, Han-Chieh Chao, Philip S. Yu". |
| | So it appears that JIAYI SUN has authored 2 articles based on the provided texts. |
| | Let me double-check if there are any other articles by JIAYI SUN: |
| | `<...we cut it down to save space...>` |
| | So, based on the provided articles, JIAYI SUN has authored 2 articles: 1. "Metaverse: Survey, Applications, Security, and Opportunities" 2. "Big Data Meets Metaverse: A Survey" |
| | Let me check if there's any other mention of JIAYI SUN in the text. |
| | In the "Big Data Meets Metaverse: A Survey" article, the authors are listed as "Jiayi Sun, Wensheng Gan, Zefeng Chen, Junhui Li, Philip S. Yu, Fellow, IEEE". The name is written with a space between "Jiayi" and "Sun". |
| | In the "Metaverse: Survey, Applications, Security, and Opportunities" article, the authors are listed as "JIAYI SUN, Wensheng Gan, Han-Chieh Chao, Philip S. Yu". |
| | So, I'm confident that JIAYI SUN has authored 2 articles. |
| | Actually, I just noticed that in the "Big Data Meets Metaverse: A Survey" article, the authors are listed with "Jiayi Sun" (with a space between "Jiayi" and "Sun") but in the "Metaverse: Survey, Applications, Security, and Opportunities" article, it's listed as "JIAYI SUN" (with no space between "JIAYI" and "SUN" in the text). However, these are just different ways of writing the same name - it's the same person. |
| | So the answer is 2. `</think>` |
| | I need to count how many different articles JIAYI SUN has authored based on the provided texts. |
| | Let me go through all the articles and identify those authored by JIAYI SUN: |
| | 1. "Metaverse: Survey, Applications, Security, and Opportunities" - Listed as "JIAYI SUN, Wensheng Gan, Han-Chieh Chao, Philip S. Yu" - This is one article by JIAYI SUN. |
| | 2. "Big Data Meets Metaverse: A Survey" - Listed as "Jiayi Sun, Wensheng Gan, Zefeng Chen, Junhui Li, Philip S. Yu, Fellow, IEEE" - This is another article by JIAYI SUN. |
| | I don't see any other articles authored by JIAYI SUN in the provided texts. The name appears in exactly two different articles. |
| | In the first article, the name is written as "JIAYI SUN" (with a space between the name components) and in the second article as "Jiayi Sun" (with a space between the name components). These are the same person with slightly different capitalization formatting. |
| | Therefore, JIAYI SUN has authored 2 different articles. |
| | `\boxed{2}` |

Table 18: Example reasoning traces for different SQL query templates and their corresponding questions from different models (GRPO-based Qwen2.5-7B-Instruct-1M, Qwen3-4B-Thinking-2507 and DeepSeek-R1-Distill-Llama-70B). (continued on the next page)

| DeepSeek-R1 Reasoning | Alright, I need to determine how many different articles Jiayi Sun has authored based on the provided articles. Let me go through each article one by one. |
|---|---|
| | First, the main article titled "Metaverse: Survey, Applications, Security, and Opportunities" is authored by Jiayi Sun, Wensheng Gan, Han-Chieh Chao, and Philip S. Yu. So, Jiayi Sun is one of the authors here. |
| | Next, the article "Big Data Meets Metaverse: A Survey" is authored by Jiayi Sun, Wensheng Gan, Zefeng Chen, Junhui Li, and Philip S. Yu. Again, Jiayi Sun is an author. |
| | Looking through the other articles, I see that Jiayi Sun is listed as an author in the first two articles. The other articles don't mention Jiayi Sun as an author. Therefore, Jiayi Sun has authored two distinct articles in the provided list. |
| | Wait, let me double-check. The first article is about the Metaverse, and the second is about Big Data and Metaverse. Both are distinct and authored by Jiayi Sun. The other articles are on topics like educational applications, manufacturing environments, and resource allocation, but they don't list Jiayi Sun as an author. |
| | So, the total number of different articles Jiayi Sun has authored in the provided set is two. </think> |
| | Jiayi Sun has authored the following articles: |
| | 1. "Metaverse: Survey, Applications, Security, and Opportunities" 2. "Big Data Meets Metaverse: A Survey" |
| | **Answer:** Jiayi Sun, Wensheng Gan, Han-Chieh Chao, Philip S. Yu, Zefeng Chen, Junhui Li. |
| | The number of different articles Jiayi Sun has authored is 2. |
| | \boxed{2} |

| Template 3 | `SELECT COUNT(DISTINCT article_id) FROM article_author WHERE author_name = {author-name}` |
|---|---|
| Question Context 512K | How many different articles has Qiang Cheng authored? |
| Gold Answer | 9 |

Table 18: Example reasoning traces for different SQL query templates and their corresponding questions from different models (GRPO-based Qwen2.5-7B-Instruct-1M, Qwen3-4B-Thinking-2507 and DeepSeek-R1-Distill-Llama-70B). (continued on the next page)

| GRPO Reasoning | To solve this problem, we need to follow a systematic approach by analyzing each provided article to identify instances where Qiang Cheng is listed as an author. Let's go through each step in detail. |
|---|---|
| | ### Step 1: Identify Articles Mentions Qiang Cheng as an Author |
| | First, we will examine each article title and author list to determine which articles include Qiang Cheng as an author. |
| | 1. **Article Title:** "Intelligent Reflecting Surface Enhanced Wireless Network: Joint Active and Passive Beamforming Design" |
| | - Authors listed: Qingqing Wu, Rui Zhang, \boxed{Qiang Cheng}, Tie Jun Cui. |
| | 2. **Article Title:** "Broadband continuous beam-steering with time-modulated metasurfaces in the near-infrared spectral regime" |
| | - Authors listed: Raana Sabri, Mohammad Mahdi Salary, Hossein Mosallaei, \boxed{Qiang Cheng}. |
| | 3. **Article Title:** "Path Loss Modeling and Measurements for Reconfigurable Intelligent Surfaces in the Millimeter-Wave Frequency Band" |
| | - Authors listed: Wankai Tang, Xiangyu Chen, Ming Zheng Chen, Jun Yan Dai, Yu Han, Marco Di Renzo, Shi Jin, \boxed{Qiang Cheng}, Tie Jun Cui. |
| | 4. **Article Title:** "Index Modulation for 6G Wireless Networks Using Time-Varying Metasurfaces" |
| | - Authors listed: John A. Hodge, Kumar Vijay Mishra, Amir I. Zaghloul, \boxed{Qiang Cheng}. |
| | From this analysis, we see that Qiang Cheng is an author in four different articles. |
| | ### Step 2: Count the Number of Articles Featuring Qiang Cheng |
| | Now that we have identified the articles where Qiang Cheng is an author, we can count them. The list of articles includes: |
| | 1. "Intelligent Reflecting Surface Enhanced Wireless Network: Joint Active and Passive Beamforming Design" |
| | 2. "Broadband continuous beam-steering with time-modulated metasurfaces in the near-infrared spectral regime" |
| | 3. "Path Loss Modeling and Measurements for Reconfigurable Intelligent Surfaces in the Millimeter-Wave Frequency Band" |
| | 4. "Index Modulation for 6G Wireless Networks Using Time-Varying Metasurfaces" |
| | Thus, Qiang Cheng has authored \boxed{4} different articles. |
| | ### Final Answer |
| | The number of different articles Qiang Cheng has authored is \boxed{4}. |

| Template 3 | `SELECT COUNT(DISTINCT article_id) FROM article_author WHERE author_name = {author-name}` |
|---|---|
| Question Context 1M | How many different articles has A. N. Pettitt contributed to? |
| Gold Answer | 1 |
| GRPO Reasoning | To answer the question, we will follow the steps below. |

Table 18: Example reasoning traces for different SQL query templates and their corresponding questions from different models (GRPO-based Qwen2.5-7B-Instruct-1M, Qwen3-4B-Thinking-2507 and DeepSeek-R1-Distill-Llama-70B).

