# OpenReview forum: "Who Gets Cited Most? Benchmarking Long-Context Language Models on Scientific Articles"
_ICLR.cc/2026/Conference — ICLR 2026 Conference Withdrawn Submission_

### Official Review · Reviewer_1uzy · 2025-10-20

**Soundness:** 3
**Presentation:** 3
**Contribution:** 2
**Rating:** 4
**Confidence:** 4

**Summary:**

This paper introduces SciTrek, a benchmark that evaluates LLM performance on context intensive tasks. The benchmark is formed as follows: first, a database of scientific articles is collected. Then, SQL queries like "How many references do articles with exactly 2 articles have" are used to generate questions which are translated to natural language. At evaluation, the question and the database of articles is fed as input to the LLM. The evaluation shows that LLMs do not perform well on this task (especially as the input size grows), and that finetuning them only improves performance by a small amount.

**Strengths:**

- The paper is clear and easy to follow
- The evaluation is comprehensive. Not only are a plurality of LLMs evaluated, but some are also fine-tuned for the task
- The analysis section is thorough, showing failure modes in an easy to understand manner

**Weaknesses:**

My main concern is that the benchmark seems to function somewhat like an "adversarial stress test" on model context. The setup involves loading many papers into the context and asking the model to retrieve specific information. In that sense, it feels closer to testing large-context retrieval, similar in spirit to giving the model pairs of 64k-token-long-numbers and asking the model to sum them up, rather than evaluating structured reasoning. While context understanding is a crucial tool for LLMs to have, it is ultimately a means to an end, rather than goal itself (i.e., solving problems at a human expert level or higher). I am not fully convinced that benchmarking it in isolation provides enough of a contribution as it stands.

In my view, a more convincing context usage benchmark would be one where long-context handling is genuinely necessary and where the problem cannot be solved by using alternative tools such as running code or SQL queries on the database.

**Questions:**

1. Could the authors clarify to what extent the benchmark tasks require reasoning beyond long-context retrieval? Are there examples where correct retrieval alone is insufficient, and some structured reasoning necessary?
2. How would the evaluation change if LLMs recieved the input but broken down into smaller chunks? For example, what if the papers weren't all given to the LLM at once, but in 32k token chunks, wherein the LLM could either manually query or automatically receive the next chunk when it chooses to.
3. How would performance change if LLMs were allowed to query the database using SQL?

---

> ### Author Response · Authors · 2025-11-22
> **Thanks for your comments. Our responses to weaknesses are as follows.**
>
> ***Weakness 1***:
>
> *My main concern is that the benchmark seems to function somewhat like an "adversarial stress test" on model context. The setup involves loading many papers into the context and asking the model to retrieve specific information. In that sense, it feels closer to testing large-context retrieval, similar in spirit to giving the model pairs of 64k-token-long-numbers and asking the model to sum them up, rather than evaluating structured reasoning. While context understanding is a crucial tool for LLMs to have, it is ultimately a means to an end, rather than goal itself (i.e., solving problems at a human expert level or higher). I am not fully convinced that benchmarking it in isolation provides enough of a contribution as it stands.*
>
> **Our reponse**:
>
> We focus on benchmarking long-context language models, given their capacity to process very long inputs (up to 10M tokens; Table 12) and their potential usefulness for applications such as [OpenAI Deep Research](https://openai.com/index/introducing-deep-research/) and [Google NotebookLM](https://notebooklm.google/).  We believe it’s critical to understand the long-context comprehension capabilities of these models before assuming they are capable of the complex reasoning over multiple documents required for real-world tasks. Existing benchmarks predominantly assess retrieval, whereas ours evaluates complex reasoning and information integration across entire contexts, including filtering, comparison, sorting, and aggregation operations. For example, the question "Which article has the most references?" cannot be answered through direct retrieval; instead, models must locate all titles and reference sections, count references for each article, perform cross-document comparison, and identify the article with the maximum count.
>
> ***Weakness 2***:
>
> *In my view, a more convincing context usage benchmark would be one where long-context handling is genuinely necessary and where the problem cannot be solved by using alternative tools such as running code or SQL queries on the database.*
>
> **Our response**:
>
> We exclusively use the databases based on metadata from Semantic Scholar for data construction, and during testing on full-article contexts, models cannot access the databases.
>
> Models must answer the questions based only on the textual representations of the articles. When given multiple full-text articles, models need to internally retrieve relevant information from the context and reason over the retrieved information to arrive at the final answer. This mimics a realistic scientific research workflow requiring understanding multiple scientific articles, where the long-context handling is the natural approach over running code or SQL.

---

> ### Author Response · Authors · 2025-11-22
> **Thanks for your comments. Our responses to questions are as follows.**
>
> ***Question 1***:
>
> *Could the authors clarify to what extent the benchmark tasks require reasoning beyond long-context retrieval? Are there examples where correct retrieval alone is insufficient, and some structured reasoning necessary?*
>
> **Our response**:
>
> Our benchmark heavily requires reasoning, and this is the most important difference from existing long-context benchmarks. For example, the question based on citation relationships, “How many articles have been cited by other articles but do not cite any other articles?” shown in Table 1 requires models to understand citation relationships among given articles. Models need to retrieve relevant information and then filter and count to get the final correct answer. Table 1 in our paper shows representative questions in different reasoning skills, and they all require retrieval and precise reasoning. Moreover, our results in Table 7 in our paper show that most open-source models struggle to reason over the database tables, although they don’t need to retrieve and count based on long contexts. This reveals that not only retrieval but also reasoning is challenging to most current models.
>
> ***Question 2***:
>
> *How would the evaluation change if LLMs recieved the input but broken down into smaller chunks? For example, what if the papers weren't all given to the LLM at once, but in 32k token chunks, wherein the LLM could either manually query or automatically receive the next chunk when it chooses to.*
>
> **Our response**:
>
> Our goal is to benchmark long-context comprehension, rather than approaches that reduce the context length. The token-chunking approach proposed is an interesting idea.  However, as many of our questions necessitate cross-article reasoning, we believe such a method is non-trivial and requires additional details like shared memory or multiple stages. We leave an exploration of these ideas to future work.
>
> ***Question 3***:
>
> *How would performance change if LLMs were allowed to query the database using SQL?*
>
> **Our response**:
>
> As shown in Table 7, we find that the language models perform substantially better when given the textual representations of metadata tables (not the full-article context), as this setting is inherently simpler because models do not need to retrieve or count titles, authors, or references. We would expect that if models were allowed to query the database directly, they might perform even higher.  However, as mentioned above, in realistic scenarios (such as conducting a literature review),  structured databases are not available, and models must interpret and reason directly over full-text articles as required by our benchmark.

---

> > ### Comment · Reviewer_1uzy · 2025-11-23
> >
> > Thank you for your response.
> >
> > I'm still struggling to understand the setting in which an LLM is given 1M tokens worth of papers or paper metadata. Surely in such a setting, either 1) the LLM would have access to the database, in which case it could write an SQL query, or 2) if the database is not available, it could use Google or Semantic Scholar's API and write the appropriate query.
> >
> > To me, it seems like the task is made challenging only because of this limitation (writing code isn't allowed, all context goes into the LLM at once). If I understand correctly, we can see this in Table 7; when given metadata instead of full articles, performance improves significantly.
> >
> > I agree with the authors regarding LLM context. Claims about very large context lengths often seem overstated. At the same time, I am not fully convinced about the value of benchmarking context length in isolation. It might be more informative to identify a naturally occurring and genuinely challenging task where effective reasoning over a very large context is essential.

---

> > > ### Author Response · Authors · 2025-11-27
> > >
> > > Thank you for the comments! We agree that the task is predominantly challenging due to the long-context formulation: as you noted, when given only the metadata, performance improves significantly (although it remains not perfect). The primary goal of our benchmark is to evaluate model capabilities that are fundamental to long-context language models within this long-context regime,  rather than running SQL queries or tools to do the task.
> > >
> > > As alluded to in your responses, many model releases advertise their capability to process extremely long inputs.
> > >
> > > For example, from [OpenAI’s report on GPT-4.1](https://openai.com/index/gpt-4-1/):
> > > *GPT‑4.1, GPT‑4.1 mini, and GPT‑4.1 nano can process up to 1 million tokens of context—up from 128,000 for previous GPT‑4o models. 1 million tokens is more than 8 copies of the entire React codebase, so long context is a great fit for processing large codebases, or lots of long documents.*
> > >
> > > And from [Gemini 2.5’s technical report](https://arxiv.org/abs/2507.06261):
> > > *Both Gemini 2.5 Pro and Gemini 2.5 Flash can process pieces of long-form text (such as the entirety of “Moby Dick” or “Don Quixote”), whole codebases, and long form audio and video data.*
> > >
> > > They evaluate these claims using needle-in-a-haystack (NIAH) performance. Notably, NIAH (or MRCR-V2 in the mentioned reports) is also a task that can be finished by writing a short program or loading the keys into a database and querying it directly in practice. Furthermore, NIAH does not represent any typical user’s input to an LLM or a common task. However, NIAH is being used as a proxy to gauge the model’s capability to retrieve precise information over long contexts. This simple retrieval capability may show up implicitly in many tasks, and thus is a useful capability to benchmark.
> > >
> > > Similarly, although tool calls or code-writing might be an easier solution to the questions in SciTrek, our goal is to provide a comprehensible (via our skill breakdown in Tables 3 and 4) and scalable (due to low human effort needed to create new datapoints) benchmark that directly measures capabilities implicit to real-world tasks. This is important because many realistic questions cannot be easily answered by just running SQL queries on databases or Google or Semantic Scholar's APIs. For example, one of the OpenScholar’s expert-written literature review questions on multiple scientific articles is: "Which downstream task can be solved by AlphaFold3 but cannot be performed by ESM-3?". This question implicitly requires logical operations and filtering over a retrieved set of articles, but cannot be easily answered with code or SQL. By measuring performance on our well-defined set of skills (i.e., sorting, comparing,  filtering, and aggregating), we aim to identify shortcomings in current model capabilities that are necessary for more realistic tasks. Moreover, QMDR[1] and MoNaCo[2] show that our targeted capabilities are fundamental to real-world, complex tasks across multiple documents, as evidenced by their manual decomposition of complex questions into atomic, simple questions.
> > >
> > > Also note that compared to other ‘natural/realistic’ benchmarks (Table 2), SciTrek is more scalable and supports detailed error analysis on model capabilities (Figure 2), which we argue is a better basis for model evaluation. For example, we were able to scale our analysis up to 1M tokens and show that models struggle with interleaved reasoning and retrieval. Moreover, as shown in existing benchmark studies in our related work section, creating a long-context benchmark for a naturally occurring and genuinely challenging task with minimal human effort is extremely difficult. This challenge motivates our proposed automated data-construction approach for generating benchmark questions and answers.
> > >
> > > [1] Tomer Wolfson, Mor Geva, Ankit Gupta, Yoav Goldberg, Matt Gardner, Daniel Deutch, Jonathan Berant: Break It Down: A Question Understanding Benchmark. Trans. Assoc. Comput. Linguistics 8: 183-198 (2020)
> > >
> > > [2] Tomer Wolfson, Harsh Trivedi, Mor Geva, Yoav Goldberg, Dan Roth, Tushar Khot, Ashish Sabharwal, Reut Tsarfaty: MoNaCo: More Natural and Complex Questions for Reasoning Across Dozens of Documents. CoRR abs/2508.11133 (2025)

---

### Official Review · Reviewer_Bdtq · 2025-10-26

**Soundness:** 3
**Presentation:** 3
**Contribution:** 2
**Rating:** 6
**Confidence:** 4

**Summary:**

This paper defines a benchmark for evaluating the long context performance of LLMs on questions that resemble (and are based on) SQL queries. To construct this benchmark, they gather articles from Semantic Scholar along with their metadata and construct SQL queries over this metadata by combining different operations and conditions. They then use an LLM to convert each query into natural text and provide the model with the full article texts along with this query. The performance of the model is quantified by its EM and F1 with the SQL query result as the ground-truth. In addition to providing a benchmark, the authors fine-tune a model (using SFT and GRPO) on instances from the distribution of their benchmark and illustrate that performance can be substantially improved with task-specific training.

**Strengths:**

- This paper contributes understanding of an important limitation of LLMs: their inability to properly process information from long contexts.
- The benchmark provides thorough evaluations at different levels of context and a “Database Tables” evaluation which measures the difficulty of the task without long context (i.e., when the information is presented as succinctly as possible).
- The authors offer additional insights by showing that this long context capability can be improved with task-specific training.

**Weaknesses:**

- The design of the contexts for the benchmark feels a bit unnatural.
    - The full article text is provided in the context, but the vast majority of this information is unused. The only parts that are used are metadata which is likely present in the article header. So, this feels a lot like a needle-in-the-haystack evaluation where the model needs to ignore most of the information present. It would be interesting to see whether the additional task complexity add signal over existing simpler needle-in-the-haystack evaluations.
    - The number of instances that the model is expected to perform aggregation over feels high (in the case of 1M context length, there are 46.3 articles on average). Asking a model to, for example, compute the average of ~50 values is somewhat of an unnatural task (a modern LLM or a human performing this task would use a tool for such and operation). It’s unclear whether we should expect models to be good at this, and indeed, performance is low even in the “Database Tables” setting where the information is presented succinctly. So the failures seem to be not just from long context but with difficulty of an unnatural task.
- It is unclear what insights can be derived from this evaluation beyond existing long context evaluation. Is model performance here well-correlated or uncorrelated with existing benchmarks?

**Questions:**

NA

---

> ### Author Response · Authors · 2025-11-22
> **Thank you for the detailed feedback**
>
> Thank you for the detailed feedback. In response:
>
> ***Weakness 1.1***:
>
> *The full article text is provided in the context, but the vast majority of this information is unused. […] It would be interesting to see whether the additional task complexity add signal over existing simpler needle-in-the-haystack evaluations.*
>
> **Our response**:
>
> Although our questions focus on titles, authors, and references, they do require models to reason over multiple article elements (through filtering, comparing, sorting, and aggregating). We deliberately target these core components because they represent a logical prerequisite: **models that cannot reason effectively about basic article structure cannot be expected to perform complex analysis over scientific content**.
>
> Unlike needle-in-the-haystack benchmarks that measure only retrieval, ours evaluates reasoning, which is a critical capability for complex tasks. For instance, answering "Which article has the most references?" requires multi-step reasoning: locating all titles and reference sections, counting references per article, comparing counts across articles, and identifying the maximum. This cannot be accomplished through direct retrieval alone.
>
> Our experimental results in Table 7 show that all frontier long-context language models struggle with our benchmark, while they show near-perfect performance in needle-in-the-haystack tasks (see results reported in [Qwen2.5-1M Technical Report](https://arxiv.org/abs/2501.15383)). The benchmark is thus intended to be complementary, not duplicative.
>
> ***Weakness 1.2***:
>
> *The number of instances that the model is expected to perform aggregation over feels high (in the case of 1M context length, there are 46.3 articles on average). […] So the failures seem to be not just from long context but with difficulty of an unnatural task.*
>
> **Our response**:
>
> We aim to benchmark long-context language models that claim to support inputs up to 10M tokens (Table 12). While our questions may seem unnatural for humans, they allow us to scalably evaluate fundamental capabilities of language models to reason over long contexts. We hypothesise that these capabilities are necessary for complex real-world tasks in scientific domains.
>
> ***Weakness 2***:
>
> *It is unclear what insights can be derived from this evaluation beyond existing long context evaluation. Is model performance here well-correlated or uncorrelated with existing benchmarks?*
>
> **Our response**:
>
> Existing benchmarks focus predominantly on retrieval, while ours evaluates interleaved reasoning and retrieval. As mentioned above, frontier models exhibit near-perfect needle-in-the-haystack benchmark performance, but perform poorly on our benchmark.
>
> With regard to insights, our failure analysis in Table 9 reveals an important distinction: failures in most models correlate with question length, whereas the best-performing models (o4-mini, Qwen3-4B-Thinking-2507, and Qwen3-30B-A3B-Thinking-2507) show stronger correlation with the complexity of underlying SQL queries. This suggests that these models not only struggle with handling the long context but also with performing complex reasoning in this domain.

---

### Official Review · Reviewer_iVac · 2025-10-27

**Soundness:** 3
**Presentation:** 1
**Contribution:** 2
**Rating:** 4
**Confidence:** 4

**Summary:**

The paper introduces SciTrek, a benchmark designed to evaluate the long-context capabilities of large language models (LLMs) in scientific domains. SciTrek employs an automatic pipeline that converts article-metadata into a relational database and then generates natural language questions from SQL queries, offering explicit, fine-grained reasoning steps and clear ground-truth answers. Moreover, the benchmark is scalable, supporting context lengths up to 1M tokens and minimal human supervision in generation. While the framework aims to capture the importance of scientific reasoning tasks, its design primarily emphasizes verifiable outputs through SQL-based queries, which may limit its ability to fully reflect naturalistic or realistic scientific reasoning processes.

**Strengths:**

- The use of SQL queries allows verifiable and objective evaluation of model outputs.
- Inter-annotator agreement is measured, strengthening the reliability of annotations.
- The benchmark demonstrates scalability and flexible context length control, offering potential adaptability for different experimental settings.

**Weaknesses:**

- Although LCLMs are highly relevant for scientific tasks, the benchmark's question design does not reflect real-world scientific use cases. Transforming SQL queries into questions does not inherently yield realistic or natural contexts.
- The input construction, based on concatenated papers, is unlikely to mirror authentic scientific workflows or user prompts, suggesting the dataset is closer to an SQL style long context benchmark rather than a naturalistic scientific one.
- Relevance through citation links does not guarantee realistic contextual usage, as the dataset lacks evidence that article groupings represent actual scenarios.
- The benchmark is limited to numeric operation tasks, which capture only a narrow slice of long-context reasoning capabilities. Consequently, it fails to assess diverse cognitive demands across scientific reasoning dimensions.
- The tasks depth and domain diversity are insufficient to justify the claim of comprehensively assessing long-context capabilities in scientific literature, especially in light of recent work such as ETHIC [1] and Is it really long context if all you need is retrieval? [2].
- Table 2 raises concerns regarding the classification of skills (e.g., why Loong [3] does not require multiple reasoning capabilities?), and the claim of "natural" benchmarking appears overstated given the benchmark's artificial construction.
----
*[1] ETHIC: Evaluating Large Language Models on Long-Context Tasks with High Information Coverage*
*[2] Is it really long context if all you need is retrieval? towards genuinely difficult long context nlp*
*[3] Leave No Document Behind: Benchmarking Long-Context LLMs with Extended Multi-Doc QA*

**Questions:**

- In table 7, the distinction between full-text articles and database tables is unclear. Why can the latter handle 512k or 1M contexts when the former cannot-does this reflect genuine context length handling or merely reduced tokenization cost?
- Citation order inconsistencies should be standardized throughout (e.g., Dubey et al., 2024; Kamath et al., 2025; Yang et al., 2025; Llama Team, 2025 on page 3 is repeated in the exact reverse order on the same page as Llama Team, 2025; Yang et al., 2025; Kamath et al., 2025; Dubey et al., 2024).
- Repeated mention of OpenScholar and LongBench v2 across sections could be condensed.
- Minor typographical issues (e.g., Semantic Scholar's footnote 6 on page 5 is different with other footnote cases).
- Figure 2 suffers from low readability. Improved visual clarity would enhance presentation quality.
- The manuscript mentions evaluation of the model's reasoning process, yet no explicit metric or mechanism for such assessment is described. Without this, the benchmark becomes indistinguishable from prior long-context evaluation datasets.

---

> ### Author Response · Authors · 2025-11-22
> **Thanks for your comments. Our responses to weaknesses are as follows.**
>
> ***Weakness 1**:*
>
> *Although LCLMs are highly relevant for scientific tasks, the benchmark's question design does not reflect real-world scientific use cases. [...]*
>
> **Our response**:
>
> Even though the questions themselves are relatively superficial (focused only on titles, authors, and references in scientific articles), reasoning over these elements is fundamental for any realistic understanding of multiple scientific articles. Notably, our questions require models to retrieve and reason over long contexts, performing operations such as filtering, comparing, sorting, and aggregating. It may be impossible for models to perform complex comprehension if they cannot first reason effectively about the core components of scientific articles.
>
> Our question generation procedure requires minimal human effort, whereas more realistic questions (e.g. those targeting domain-specific scientific content) depend heavily on manual question and answer creation.
>
> ***Weakness 2**:*
>
> *The input construction, based on concatenated papers, is unlikely to mirror authentic scientific workflows or user prompts, [...]*
>
> **Our response**:
>
> Our aim was not to emulate full scientific workflows but to probe the reasoning capabilities of long-context language models. We agree that concatenating papers does not perfectly reflect how scientists typically interact with literature; however, we hypothesise that our concatenated-paper context would be useful for Deep Research-style products such as [OpenAI Deep Research](https://openai.com/index/introducing-deep-research/) and [Google NotebookLM](https://notebooklm.google/). Thus, there is an urgent need to rigorously evaluate the reasoning capabilities of frontier long-context language models.
>
> ***Weakness 3***:
>
> *Relevance through citation links does not guarantee realistic contextual usage, [...].*
>
> **Our response**:
>
> To streamline context construction, we create groups either by sampling random articles from diverse domains or by selecting articles connected through citation chains, which typically remain within the same domain. Random groupings expose models to heterogeneous, cross-domain contexts, while citation-based groupings introduce richer relational structure that demands more advanced long-context reasoning. The reasoning skills evaluated in our benchmark, filtering, comparing, sorting, and aggregating based on citation relationships, are foundational for realistic contextual use.
>
> ***Weakness 4***:
>
> *The benchmark is limited to numeric operation tasks, which capture only a narrow slice of long-context reasoning capabilities [...]*
>
> **Our response**:
>
> Our goal is to provide a rigorous evaluation of long-context models on focused reasoning tasks (filtering, comparing, sorting, and aggregating information from scientific articles) rather than to capture the full diversity of scientific reasoning.
> Table 7 demonstrates that existing long-context models struggle even with these core skills, achieving under 50% exact-match accuracy. Given the difficulty of automatically creating reliable scientific reasoning questions and answers, our SQL-driven simulation framework represents a key contribution, enabling automated benchmark construction with minimal human involvement.
>
> ***Weakness 5***:
>
> *The tasks depth and domain diversity are insufficient to justify the claim of comprehensively assessing long-context capabilities in scientific literature, especially in light of recent work such as ETHIC [1] and Is it really long context if all you need is retrieval? [2].*
>
> **Our response**:
>
> We moderate our claim by noting that we do not aim to assess all long-context capabilities relevant to scientific literature. Instead, our focus is on automatically generating questions and answers to benchmark specific reasoning capabilities of long-context language models. Our benchmark targets reasoning over core components of scientific articles (titles, authors, and references) on the premise that advanced scientific reasoning is unlikely if models cannot first handle these simple elements.
>
> Unlike [1] and [2], which do not consider reasoning difficulty or structure, we explicitly consider the underlying reasoning processes required to answer each question and formalise these processes through a reasoning typology grounded in SQL operations over databases.
>
> ***Weakness 6***:
>
> *Table 2 raises concerns regarding the classification of skills (e.g., why Loong [3] does not require multiple reasoning capabilities?), and the claim of "natural" benchmarking appears overstated given the benchmark's artificial construction.*
>
> **Our response**:
>
> The term ‘skill’ in Table 2 refers to the diversity of question types. We distinguish our benchmark from Loong based on the diversity of skills. Loong only uses exactly two question templates for their multi-document tasks on scientific articles, while we have questions based on 387 different SQL query templates covering 6 high-level capabilities shown in Table 4.

---

> ### Author Response · Authors · 2025-11-22
> **Thanks for your comments. Our responses to questions are as follows.**
>
> ***Question 1***:
>
> *In table 7, the distinction between full-text articles and database tables is unclear. Why can the latter handle 512k or 1M contexts when the former cannot-does this reflect genuine context length handling or merely reduced tokenization cost?*
>
> **Our response**:
>
> To clarify, we have two experimental testing settings with different inputs: full-text articles and the textual representation of corresponding database tables. The full-text articles measure the models’ ability to retrieve and reason over long contexts, while the database setting measures just the reasoning capabilities without long-context retrieval. The database tables and their textual representations are further described in Appendix A in our submission. As mentioned in the caption, although the full-article contexts reach 1M tokens, textual representations of the corresponding database contexts are only about 2K tokens long on average.
>
> ***Question 2***:
>
> *Citation order inconsistencies should be standardized throughout.*
>
> **Our response**:
>
> Updated in the new version of the submission.
>
> ***Question 3***:
>
> *Repeated mention of OpenScholar and LongBench v2 across sections could be condensed.*
> *Minor typographical issues [...]*
>
> **Our response**:
>
> Updated in the new version of the submission.
>
> ***Question 4**:*
>
> *Figure 2 suffers from low readability.*
>
> **Our response**:
>
> We have updated the models included in the figure and the figure caption in the new version of our submission.
>
> ***Question 5***:
>
> *The manuscript mentions evaluation of the model's reasoning process, yet no explicit metric or mechanism for such assessment is described. Without this, the benchmark becomes indistinguishable from prior long-context evaluation datasets.*
>
> **Our response**:
>
> There is some misunderstanding. Our dataset creation methodology allows us to analyse the model’s reasoning process via its answer quality as we have an explicit reasoning typology for our questions (i.e., filtering, comparing, sorting, and aggregating). For example, Figure 2 in our paper shows Deepseek-R1-Distill-Llama-70B performs poorly on questions requiring sorting capabilities. Therefore, we eliminate the need to directly measure the quality of the reasoning traces.
> While we focus on reasoning, the existing benchmarks mostly consider information retrieval. Their answers can be directly retrieved from the contexts, but our answers can only be derived by reasoning from the inputs. This poses challenges to the current long-context language models based on our experimental results in Table 7, although they perform well in information retrieval, such as needle-in-the-haystack tasks (see results reported in [Qwen2.5-1M Technical Report](https://arxiv.org/abs/2501.15383)).

---

> > ### Comment · Reviewer_iVac · 2025-11-24
> > **Official Comment by Reviewer iVac**
> >
> > Thank you for the detailed rebuttal and the updated version of the paper. I increased the presentation score. However, I still have several remaining concerns:
> >
> > ---
> >
> > - You describe your questions as “relatively superficial” and confined to titles, authors, and references, yet also call the associated operations “fundamental” and suggest that advanced scientific reasoning is unlikely without them. Please ground this claim more clearly in prior work (e.g., citation analysis, metadata-centric tools) and spell out which concrete scientific workflows your six information-processing skills approximate, so that SciTrek is clearly understood as more than a narrow test of numerical operations over metadata.
> >
> > - You motivate SciTrek via Deep-Research-style systems and suggest that concatenated-paper contexts would be useful for such products, but these tools use multi-hop, query-driven retrieval rather than a single fixed bundle of papers, whereas SciTrek groups papers randomly or via citations and keeps that group fixed. If you keep this analogy, please clarify which aspects of agentic, multi-step literature review your setting captures and which (e.g., query-driven expansion beyond immediate citations) are explicitly out of scope, so as not to overstate the connection to real “deep research” workflows.
> >
> > - The paper and rebuttal now clearly state that SciTrek focuses on basic numerical operations (counting, aggregating, sorting) over metadata and does not aim to cover all long-context capabilities in scientific literature. My concern is that parts of the narrative still invite readers to treat SciTrek performance as a strong indicator of scientific long-context reasoning, even though many dimensions (e.g., understanding methods, interpreting figures, multi-hop content reasoning) are out of scope and are also targeted by benchmarks such as ETHIC and HELMET, so I would encourage you to frame SciTrek explicitly as a benchmark for structured metadata-level reasoning in long contexts and to list these missing dimensions as limitations.
> >
> > - In Table 2 and the rebuttal you contrast SciTrek with Loong by stating that Loong “only uses exactly two question templates for their multi-document tasks on scientific articles,” whereas SciTrek has 387 SQL templates covering six skills. However, Loong defines four main task categories (Spotlight Locating, Comparison, Clustering, Chain of Reasoning) and several sub-tasks over multi-document inputs, while in your work the 387 templates instantiate six information-processing skills rather than 387 distinct capabilities, so it would help to state explicitly that “skills” in Table 2 refers to your six SQL-defined operations (not the template count) and to revise the description of Loong’s templates so that it aligns with their task categories.
> >
> > - Table 2 marks SciTrek as “Natural,” and Section 1 rightly emphasises that you work with real scientific articles rather than synthetic contexts, but you also describe the questions as “relatively superficial” and note that you are not emulating full scientific workflows or more realistic, domain-specific questions. In this light, the “Natural” label can be ambiguous—some readers may take it to mean naturalistic tasks or questions rather than natural-source documents—so clarifying that “Natural” refers to using real-world documents (and, if helpful, renaming the column to something like “Natural text” or “Real documents”) would better reflect your intent.
> >
> > - Your rebuttal to Question 1 clarifies that full-text contexts can reach 1M tokens whereas the corresponding database-table inputs are only about 2K tokens on average. Even so, placing the database-table results under the same 64K/128K/512K/1M columns makes it easy to misread them as 64K/128K/512K/1M-token DB inputs, so you might either separate the database-table results into their own table keyed by collection group or add a column with actual average token lengths, so that the 64K/128K/512K/1M labels are clearly interpreted as properties of the underlying full-text collections.
> >
> > - In response to Question 5, you argue that associating each question with a specific skill type (filtering, comparing, sorting, aggregating, etc.) lets you “analyse the model’s reasoning process via its answer quality,” and highlight explicit SQL-based reasoning steps. While the SQL backbone makes the intended reasoning structure explicit and supports detailed error analysis, the evaluation still uses exact match/F1 on final answers with reasoning traces only qualitatively inspected for one GRPO model, so this seems closer to indirect assessment of reasoning via performance on typed tasks, and I would suggest softening claims that SciTrek “evaluates” reasoning processes and instead emphasising that it provides an explicit, automatically generated labelling of reasoning-type categories.
> >
> > ---
> >
> > Following those concerns, I have decided to keep my overall scores. Thank you again for your detailed response.

---

> > > ### Author Response · Authors · 2025-11-27
> > > **Thanks for the comments. These are our responses to the concerns below. (Part 2/3)**
> > >
> > > ***Concern 3**:*
> > >
> > > *The paper and rebuttal now clearly state that SciTrek focuses on basic numerical operations (counting, aggregating, sorting) over metadata and does not aim to cover all long-context capabilities in scientific literature. My concern is that parts of the narrative still invite readers to treat SciTrek performance as a strong indicator of scientific long-context reasoning, even though many dimensions (e.g., understanding methods, interpreting figures, multi-hop content reasoning) are out of scope and are also targeted by benchmarks such as ETHIC and HELMET, so I would encourage you to frame SciTrek explicitly as a benchmark for structured metadata-level reasoning in long contexts and to list these missing dimensions as limitations.*
> > >
> > > **Our response**:
> > >
> > > We stated in the introduction (line 095) that our benchmark covers fundamental reasoning capabilities of long-context language models over scientific articles while our created questions are focused on titles, authors, and references. As explained in the response to Concern 1, our focused reasoning skills are implicitly required in realistic scientific tasks. We have added more limitations on the missing dimensions of scientific long-context reasoning.
> > >
> > >
> > > ***Concern 4**:*
> > >
> > >  *In Table 2 and the rebuttal you contrast SciTrek with Loong by stating that Loong “only uses exactly two question templates for their multi-document tasks on scientific articles,” whereas SciTrek has 387 SQL templates covering six skills. However, Loong defines four main task categories (Spotlight Locating, Comparison, Clustering, Chain of Reasoning) and several sub-tasks over multi-document inputs, while in your work the 387 templates instantiate six information-processing skills rather than 387 distinct capabilities, so it would help to state explicitly that “skills” in Table 2 refers to your six SQL-defined operations (not the template count) and to revise the description of Loong’s templates so that it aligns with their task categories.*
> > >
> > > **Our response**:
> > >
> > > We have changed the column of “Skills” to “#SciQs” in Table 2, and it denotes the number of unique benchmark questions or instructions with gold answers on scientific articles. Loong presents high-quality data with multiple skills across domains for benchmarking long-context language models, and it has 53 unique questions based on scientific articles. In SciTrek, there are 1,716 unique questions associated with specific information-processing skills based on different underlying SQL queries (Tables 3 and 4). This allows more detailed error analysis on reasoning capabilities of long-context language models over scientific articles. We have made correponding updates in the table caption of Table 2 in the new version of our submission.
> > >
> > >
> > > ***Concern 5**:*
> > >
> > > *Table 2 marks SciTrek as “Natural,” and Section 1 rightly emphasises that you work with real scientific articles rather than synthetic contexts, but you also describe the questions as “relatively superficial” and note that you are not emulating full scientific workflows or more realistic, domain-specific questions. In this light, the “Natural” label can be ambiguous—some readers may take it to mean naturalistic tasks or questions rather than natural-source documents—so clarifying that “Natural” refers to using real-world documents (and, if helpful, renaming the column to something like “Natural text” or “Real documents”) would better reflect your intent.*
> > >
> > > **Our response**:
> > >
> > > We have updated the column title to “Natural Context” now, although we had a clarification on “Natural” in the table caption in the previous version of our submission.

---

> > > ### Author Response · Authors · 2025-11-27
> > > **Thanks for the comments. These are our responses to the concerns below. (Part 3/3)**
> > >
> > > ***Concern 6**:*
> > >
> > > *Your rebuttal to Question 1 clarifies that full-text contexts can reach 1M tokens whereas the corresponding database-table inputs are only about 2K tokens on average. Even so, placing the database-table results under the same 64K/128K/512K/1M columns makes it easy to misread them as 64K/128K/512K/1M-token DB inputs, so you might either separate the database-table results into their own table keyed by collection group or add a column with actual average token lengths, so that the 64K/128K/512K/1M labels are clearly interpreted as properties of the underlying full-text collections.*
> > >
> > > **Our response**:
> > >
> > > Thanks for the suggestions. We have updated the result tables in the new version of our submission, Tables 7 and 13.
> > >
> > > ***Concern 7**:*
> > >
> > > *In response to Question 5, you argue that associating each question with a specific skill type (filtering, comparing, sorting, aggregating, etc.) lets you “analyse the model’s reasoning process via its answer quality,” and highlight explicit SQL-based reasoning steps. While the SQL backbone makes the intended reasoning structure explicit and supports detailed error analysis, the evaluation still uses exact match/F1 on final answers with reasoning traces only qualitatively inspected for one GRPO model, so this seems closer to indirect assessment of reasoning via performance on typed tasks, and I would suggest softening claims that SciTrek “evaluates” reasoning processes and instead emphasising that it provides an explicit, automatically generated labelling of reasoning-type categories.*
> > >
> > > **Our response**:
> > >
> > > Yes, it is an indirect assessment of reasoning via performance. Our SQL-based skill types support more detailed error analysis on model capabilities. For example, based on our results in Figure 2, we can see that GPT4.1 struggles in sorting while o4-mini performs much better. We have softened our claims on the evaluation of the reasoning processes in the new version of the submission.

---

> ### Author Response · Authors · 2025-11-27
> **Thanks for the comments. These are our responses to the concerns below. (Part 1/3)**
>
> ***Concern 1**:*
>
> *You describe your questions as “relatively superficial” and confined to titles, authors, and references, yet also call the associated operations “fundamental” and suggest that advanced scientific reasoning is unlikely without them. Please ground this claim more clearly [...].*
>
> **Our response**:
>
> Our focused model capabilities are based on atomic SQL commands (i.e., sorting, filtering, and aggregating) and operators (i.e., comparison, arithmetic and logical), shown in Tables 3 and 4. We build a benchmark on these capabilities as they are explicitly required in realistic scientific tasks. For example, expert-written literature review questions from OpenScholar implicitly require these capabilities. There are five example questions from their released data and corresponding capabilities they require.
>
> - Example Question 1: "What are the latest works on finetuning an auto-regressive LM for dense passage retrieval? How are their performance compared with bi-directional encoders?"
> (Required Fundamental Capabilities: sorting, arithmetic, logical)
>
> - Example Question 2: "Which downstream task can solved by AlphaFold3 but cannot performed by ESM-3?"
> (Required Fundamental Capabilities: filtering, logical)
>
> - Example Question 3: "Citation graph is a good tool to find relevant work and to help understand the evolution of the domain. What recent research has been done to analyze scientific discourse and generate citation content using citation networks and related textual information?"
> (Required Fundamental Capabilities: sorting)
>
> - Example Question 4: "What types of mechanical resonators have been used to couple with superconducting qubits?"
> (Required Fundamental Capabilities: filtering)
>
> - Example Question 5: “Compared to 2023, how has the percentage of finished goods apparel factories from countries other than Vietnam, China, and Cambodia changed in 2024?”
> (Required Fundamental Capabilities: arithmetic, aggregating, comparing)
>
> These capabilites are important to real-world tasks, even in expert-written example questions for the financial domain from LongBench v2 shown in the example question below.
>
> - Example Question: “In the financial reports of Apple Inc. and Samsung Electronics for the years 2022 and 2023, which company has a higher percentage of revenue derived from the product category of phones, and in what range do the differences in this dependency between the two companies in the two years fall?”
> (Required Fundamental Capabilities: comparing, filtering, aggregating, logical, and arithmetic)
>
> While OpenScholar and LongBench v2 have more realistic questions, they do not support detailed error analysis to check where models fail as there are no labels of the required capabilities and they are not scalable because of heavy human effort in data construction. To proxy this, we build our scalable benchmark with questions and answers approximated by database simulation, and we provide an explicit label of reasoning capabilities for each question.
>
> Moreover, QMDR[1] presents a widely used formalism, in which complex questions can be decomposed into atomic, simple questions that require filtering, aggregating, comparing, sorting, logical and arithmetic operations.  A recent work, MoNaCo[2], further shows that real-world complex questions on multi-document understanding often demand aggregating and arithmetic skills based on their manual decomposition of naturally occurring human-written questions.
>
> We have added the reference papers in the introduction of the new version of our submission, and the example questions from OpenScholar and LongBench v2 in the appendix.
>
> [1] Tomer Wolfson, Mor Geva, Ankit Gupta, Yoav Goldberg, Matt Gardner, Daniel Deutch, Jonathan Berant: Break It Down: A Question Understanding Benchmark. Trans. Assoc. Comput. Linguistics 8: 183-198 (2020)
>
> [2] Tomer Wolfson, Harsh Trivedi, Mor Geva, Yoav Goldberg, Dan Roth, Tushar Khot, Ashish Sabharwal, Reut Tsarfaty: MoNaCo: More Natural and Complex Questions for Reasoning Across Dozens of Documents. CoRR abs/2508.11133 (2025)
>
> ***Concern 2**:*
>
> *You motivate SciTrek via Deep-Research-style systems and suggest that concatenated-paper contexts would be useful for such products, but these tools use multi-hop, query-driven retrieval rather than a single fixed bundle of papers, [...].*
>
> **Our response**:
>
> We agree that the Deep Research products use query-driven retrieval rather than a fixed batch of papers. Our benchmark is more focused on the understanding stage of multiple scientific articles when retrieved articles are provided, especially fundamental reasoning and information integration over the retrieved articles. To simplify the task, we use concatenated-paper contexts and create questions based on concatenated papers. Query-driven retrieval and expansion are out of the scope of our work. We have updated our introduction in the new version of the submission.

---

### Official Review · Reviewer_s42S · 2025-10-29

**Soundness:** 2
**Presentation:** 3
**Contribution:** 2
**Rating:** 4
**Confidence:** 4

**Summary:**

In this paper, the authors introduce SciTrek, a new long-context question-answering (QA) benchmark constructed from scientific papers. The work aims to address a key limitation of existing long-context QA datasets, which often rely on artificially constructed or non-scientific texts, by grounding the benchmark in authentic scientific literature. Using Semantic Scholar metadata, the authors automate the entire benchmark construction pipeline—from paper preprocessing and database creation to question and answer generation. They further present baseline results across several long-context language models and report performance improvements achieved through supervised fine-tuning (SFT) and reinforcement learning (RL).

**Strengths:**

The paper presents an automated pipeline for benchmark construction, which effectively reduces the manual cost and effort traditionally required for question generation. This automation offers a scalable and efficient alternative to labor-intensive dataset creation processes.

**Weaknesses:**

Limited benefit from using scientific papers
Although the benchmark is based on scientific literature, the actual advantage of this choice remains unclear. Compared to datasets such as LongBench-v2 or OpenScholar, most questions in SciTrek focus on factual or computational details that can be directly retrieved from the text, rather than deeper semantic or reasoning-based understanding of the scientific content. While the authors argue that previous long-context QA benchmarks rely on non-scientific texts, SciTrek itself does not appear to exploit domain-specific scientific knowledge or reasoning. The minimal performance variation across different scientific disciplines further supports this observation.

Insufficient experimental depth and analysis
The experimental setup appears somewhat limited, as the evaluated open-source models seem outdated. In the reasoning analysis, only three SQL query types and four corresponding reasoning traces were examined, which is insufficient for drawing generalized conclusions about reasoning behavior. Moreover, the analysis focuses primarily on the GRPO-based model; extending this examination to include failure cases and other model types would provide a more comprehensive and insightful understanding of the underlying reasoning challenges.

**Questions:**

Although the Qwen2.5 model achieves higher performance than GPT-4.1 after both SFT and RL training, the paper still concludes that it shows “poor performance.” Could the authors clarify the basis for this assessment? What specific criteria or aspects led to this judgment despite the overall performance improvement?

The models used in the experiments appear somewhat outdated. Have the authors conducted, or do they plan to conduct, baseline evaluations using more recent models such as Qwen3 (e.g., Qwen-70B-Instruct, which supports up to 256K context length) to provide a fairer comparison?

In the Relational Filtering skill evaluation, there is a noticeable gap between the “Agree” and “Align” metrics, suggesting that the quality of generated questions may be lower than for other skills. It seems plausible that question quality degrades as the query structure becomes more complex. Do the authors have any plans or strategies to improve data quality in such cases?

---

> ### Author Response · Authors · 2025-11-21
> **Thanks for the comments. Our responses to weaknesses are as follows.**
>
> **Weakness 1**:
>
> Limited benefit from using scientific papers. [...]
>
> **Our response**:
>
> With scientific articles, we can have natural long contexts and automatically constructed questions and answers.
>
> (1) Reading multiple related scientific articles is a real-world task in the workflow of researchers, so our long-context input is not synthetic, while most long-context benchmarks rely on synthetic contexts.
>
> (2) We can automatically create questions and answers based on database simulation on the metadata of scientific articles from Semantic Scholar, while data construction for ScholarBench and LongBench v2 heavily relies on human effort.
>
> (3) As explained in our submission (line 095), although our questions are relatively superficial (primarily focusing on titles, authors and references in scientific articles), they require models to effectively not only retrieve but also reason over long contexts (i.e., filtering, comparing, sorting, and aggregating).
>
> (4) We focus on fundamental reasoning over core article elements rather than domain-specific scientific content. This is a logical prerequisite: models must first master reasoning about basic article components before they can perform specialised domain-specific reasoning.
>
> **Weakness 2**:
>
> Insufficient experimental depth and analysis. [...]
>
> **Our response**:
>
> (1) While most of our experimental models were released within the past six months, to alleviate this concern, we added more experiments with Qwen3 4B and 30B models in both Instruct and Thinking modes. Experimental results show that these recent models still struggle to perform well in full-article contexts, and performance deteriorates as the scale of the input increases. Compared with Qwen2.5-1M models, Qwen3 models in thinking mode have much better performance. These results have been included in the revised version of our paper.
>
> Table-A: Test results on full-article contexts in terms of Exact Match (%) (— denotes that the model cannot support the corresponding input size)
>
> | Model                            | 64K  | 128K | 512K | 1M   |
> | -------------------------------- | ---- | ---- | ---- | ---- |
> | Qwen/Qwen3-30B-A3B-Thinking-2507 | 53.3 | 42.0 | —    | —    |
> | Qwen/Qwen3-30B-A3B-Instruct-2507 | 5.4  | 3.2  | —    | —    |
> | Qwen/Qwen3-4B-Thinking-2507      | 41.1 | 29.3 | —    | —    |
> | Qwen/Qwen3-4B-Instruct-2507      | 2.1  | 2.3  | —    | —    |
>
> Table-B: Test results on corresponding database-table contexts in terms of Exact Match (%)
>
> | Model                            | 64K  | 128K | 512K | 1M   |
> | -------------------------------- | ---- | ---- | ---- | ---- |
> | Qwen/Qwen3-30B-A3B-Thinking-2507 | 92.3 | 86.2 | 73.5 | 61.1 |
> | Qwen/Qwen3-30B-A3B-Instruct-2507 | 29.8 | 21.2 | 6.0  | 4.2  |
> | Qwen/Qwen3-4B-Thinking-2507      | 90.2 | 83.1 | 71.8 | 52.3 |
> | Qwen/Qwen3-4B-Instruct-2507      | 25.9 | 16.9 | 6.9  | 2.8  |
>
> (2) For the reasoning analysis, we actually analysed 100 reasoning traces from our GRPO-based model for 25 different random SQL query templates. In the Appendix, we only show 12 *example* traces for 3 SQL templates. This has been clarified in the new version of the paper.
>
> (3) To further address the concerns about limited analysis, we conducted additional analysis of 100 randomly sampled reasoning traces from DeepSeek-R1-Distill-Llama-70B and Qwen3-4B-A3B-Thinking-2507 for the same set of SQL query templates. These are popular reasoning models showing impressive performance (see Table 7 in our submission). We included their reasoning traces in Appendix I.2. Unfortunately, we cannot obtain the reasoning traces from closed-source models for analysis.
>
> Taken together, our analysis of reasoning traces gets similar findings as before. (1) The overall reasoning remains logically sound at 64K and 128K context lengths for all three models, although the model occasionally includes repeated steps or extraneous operations unrelated to the question. (2) Despite not being trained for 512K, our GRPO-based model still demonstrates coherent reasoning, but with reduced accuracy in specific steps compared to 64K and 128K. (3) Most reasoning steps that require counting are not accurate, especially for references, which leads to incorrect answers from these models, especially for our GRPO-based model.

---

> ### Author Response · Authors · 2025-11-22
> **Thanks for your comments. Our responses to questions are as follows.**
>
> ***Question 1**:*
>
> *Although the Qwen2.5 model achieves higher performance than GPT-4.1 after both SFT and RL training, the paper still concludes that it shows “poor performance.” Could the authors clarify the basis for this assessment? What specific criteria or aspects led to this judgment despite the overall performance improvement?*
>
> **Our response**:
>
> While models trained with SFT and RL perform better than GPT-4.1, their performance in general remains poor because the absolute value (22.5%) is low and still remains inferior to Gemini (28.1%) and o4-mini (48.4%).
>
> ***Question 2**:*
>
> *The models used in the experiments appear somewhat outdated. Have the authors conducted, or do they plan to conduct, baseline evaluations using more recent models such as Qwen3 (e.g., Qwen-70B-Instruct, which supports up to 256K context length) to provide a fairer comparison?*
>
> **Our response**:
>
> Please refer to the tables in the response to Weakness 2 above. We ran additional experiments on Qwen3 4B and 30B in both Instruct and Thinking modes, while we cannot run Qwen3 235B because of limited computing resources (4 NVIDIA HGX H200s).
>
> ***Question 3**:*
>
> *In the Relational Filtering skill evaluation, there is a noticeable gap between the “Agree” and “Align” metrics, suggesting that the quality of generated questions may be lower than for other skills. It seems plausible that question quality degrades as the query structure becomes more complex. Do the authors have any plans or strategies to improve data quality in such cases?*
>
> **Our response**:
>
> We manually checked all 120 questions that were given to human annotators. Amongst the Relational Filtering questions, we found 3 out of 19 were ambiguous, leading to the gap between the alignment and agreement for that category in Table 6. We used Qwen2.5-Coder-32B-Instruct to convert SQL queries to natural language questions. Although this mapping is fairly accurate, the complexity of SQL queries in Relational Filtering may introduce ambiguity in the questions. We have updated this analysis in the revised version of our submission. To prevent ambiguous questions, we could use three different models when converting natural language questions back to SQL queries. If the SQL queries yield the same output across models, it is more likely that there is no ambiguity; otherwise, we could filter out the corresponding question.

---

> > ### Comment · Reviewer_s42S · 2025-11-28
> > **Thank you for the response.**
> >
> > Thank you for the detailed rebuttal. The additional experiments and extended reasoning-trace analysis strengthen the technical depth of the submission. However, regarding Weakness 1, I still find the authors’ justification insufficient, and my original concern largely remains.
> >
> > Since the benchmark relies exclusively on metadata fields (titles, authors, reference counts, citation links), I do not see why scientific articles are necessary as the source context. Using such metadata does not meaningfully reflect the “core elements” of scientific articles, nor does it involve any scientific discourse or domain reasoning. If metadata is all that matters, any corpus with a similar metadata structure could be substituted with no impact on the task. Moreover, the use of SQL to generate questions and answers for long-context is not conceptually new, as prior work such as HoloBench has already adopted database-operation-style reasoning to evaluate long-context capabilities. In that sense, the primary distinction the authors claim is the naturalness rather than synthetic context, but it feels insufficient to establish a strong contribution, especially when the reasoning task itself remains entirely synthetic and independent of the scientific content of the documents.
> >
> > Therefore, I retain my scores.

---

### Author Response · Authors · 2025-11-21
**Thanks to all four reviewers for their thoughtful feedback**

Thanks to all four reviewers for their thoughtful feedback on our submission! As there are some shared comments/questions amongst reviewers, we address them here and provide additional details in specific responses to each reviewer.

(1) We use titles, authors, and references of scientific articles as a proxy to create reasoning-required questions and answers. These are core elements of scientific articles. It may be impossible for long-context language models to perform advanced reasoning if they cannot first reason effectively about these simple elements. We are not aiming at a benchmark covering all capabilities in the scientific workflow.

(2) There is an urgent need to rigorously evaluate rapidly evolving long-context language models as they are claimed to support even 10M tokens. However, the existing benchmarks predominantly consider information retrieval, despite reasoning being a critical capability for many real-world tasks. In contrast, our benchmark requires interleaving reasoning and retrieval across long contexts. Unlike existing benchmarks, where answers can be directly retrieved from the contexts, answers to our questions must be derived by reasoning on retrieved information.

(3) We propose an automatic data construction approach for long-context benchmarks based on databases created based on metadata from Semantic Scholar. Our target benchmark questions are based on full-article contexts, and to specifically test reasoning, we include experiments passing a textual representation of the corresponding database tables as input. In both test settings, the model cannot access the actual database tables or run SQL.

(4) Our benchmark is challenging to all frontier long-context language models, while they perform well on other long-context retrieval tasks like needle-in-a-haystack. Our results in Table 7 show that most open-source models even struggle to reason over the textual representation of database tables, despite not needing to retrieve and count from long contexts. This reveals that both retrieval and reasoning remain challenging to most current models.

We have updated our submission accordingly and highlighted changes in **blue**.

---

### Note · Authors · 2025-12-01

I have read and agree with the venue's withdrawal policy on behalf of myself and my co-authors.